

# Secondary Ice Production - No Evidence of Efficient Rime-Splintering Mechanism

Johanna S. Seidel [1,2], Alexei A. Kiselev [2], Alice Keinert [2], Frank Stratmann [1], Thomas Leisner [1], and Susan Hartmann [1]

[1]Leibniz Institute for Tropospheric Research, Departement of Atmospheric Microphysics, Leipzig, Germany
[2]Karlsruhe Institute of Technology, Institute of Meteorology and Climate Research, Karlsruhe, Germany

**Correspondence:** susan.hartmann@tropos.de

**Abstract.**

Mixed-phase clouds are essential for Earth's weather and climate system. Ice multiplication via secondary ice production (SIP) is thought to be responsible for the observed strong increase of ice particle number concentration in mixed-phase clouds. In this study, we focus on the rime-splintering also known as the Hallett-Mossop (HM) process, which still lacks physical and quantitative understanding. We report on an experimental study of rime-splintering conducted in a newly developed setup under conditions representing convective mixed-phase clouds in the temperature range of -4 °C to -10 °C. The riming process was observed with high-speed video microscopy and infrared thermography, while potential secondary ice particles (SI) in the super-micron size range were detected by a custom-build ice counter. Contrary to earlier HM experiments, where up to several hundreds of SI particles per mg rime were found at -5 °C, we found no evidence of productive SIP, which fundamentally questions the importance of rime-splintering. Further, we could exclude two potential mechanisms suggested as explanation for rime-splintering: freezing of droplets upon glancing contact with the rimer and fragmentation of spherically freezing droplets on the rimer surface. The break-off of sublimating fragile rime spires was observed to produce very few SI particles, insufficient to explain the large numbers of ice particles reported in earlier studies. In the transition regime between wet and dry growth, in analogy to phenomena of deformation of drizzle droplets upon freezing, we also observed formation of spikes on the rimer surface, which might be a source of SIP.



## 1 Introduction

Ice formation in mixed-phase clouds affects cloud radiative properties, impacts cloud electricity, precipitation formation and cloud lifetime and is therefore essential for Earth's weather and climate systems. Primary ice particles are formed by ice nucleating particles (INPs) catalyzing the nucleation process or via homogeneous freezing at temperatures below around $-38\,°C$.

In situ and remote sensing measurements of ice crystal number concentration (ICNC) in mixed-phase clouds occasionally demonstrate a strong discrepancy between the ICNC and INP concentrations of one to four orders of magnitude at moderate supercooling (Hobbs, 1969; Hobbs and Rangno, 1985; Mossop, 1985b; Hobbs and Rangno, 1990; Hogan et al., 2002; Crosier et al., 2011; Crawford et al., 2012; Heymsfield and Willis, 2014; Lawson et al., 2015; Taylor et al., 2016; Lasher-Trapp et al., 2016; Huang et al., 2017; Ladino et al., 2017; O'Shea et al., 2017; Korolev et al., 2020; Luke et al., 2021; Ramelli et al., 2021; Li et al., 2021). Such discrepancy could be explained by secondary ice production (SIP) processes increasing the total ice particle number concentration by multiplication of pre-existing ice particles (Field et al., 2017; Korolev and Leisner, 2020; Chisnell and Latham, 1976; Connolly et al., 2006; Sun et al., 2010; Crawford et al., 2012; Yano et al., 2016; Sullivan et al., 2018; Sotiropoulou et al., 2020; Georgakaki et al., 2022). Newest studies suggest that SIP is the dominant ice formation process in mixed-phase clouds (Zhao and Liu, 2021; Zhao et al., 2023). According to Korolev and Leisner (2020), SIP might proceed according to the following mechanisms: (a) droplet fragmentation during freezing (Takahashi and Yamashita, 1977; Wildeman et al., 2017; Lauber et al., 2018; Keinert et al., 2020; Kleinheins et al., 2021), (b) rime-splintering, (c) fragmentation during ice-ice particle collisions (Vardiman, 1978; Takahashi et al., 1995; Grzegorczyk et al., 2023), ice fragmentation due to (d) thermal shock (Dye and Hobbs, 1968; King and Fletcher, 1976b), (e) sublimation (Oraltay and Hallett, 1989; Dong et al., 1994; Bacon et al., 1998) and (f) activation of INPs in transient supersaturation (e.g., Prabhakaran et al., 2020). Yet another SIP mechanism occurring during the break-up of freezing droplets on impact with smaller ice particles, suggested by Phillips et al. (2018), was supported experimentally by James et al. (2021). None of these proposed SIP mechanisms has been sufficiently characterized so far.

The most widely accepted SIP mechanism is the Hallett-Mossop or more generally rime-splintering process (Hallett and Mossop, 1974; Mossop and Hallett, 1974), which suggests that the secondary ice (SI) particles are produced upon riming of a large ice particle (called rimer). Riming results from droplet–ice collisions as the ice particle falls through a cloud of supercooled droplets. Rime-splintering was identified in laboratory experiments to be active in a narrow air temperature range between $-3\,°C$ and $-8\,°C$ (Hallett and Mossop, 1974). They found a maximum SIP rate of around 350 SI particles per mg of accreted rime at near $-5\,°C$ and at a rimer velocity of $2.7\,m\,s^{-1}$. Similar results were obtained by Mossop in the follow-up experiments (Mossop, 1976, 1985a). Therefore, the temperature range from $-3\,°C$ to $-8\,°C$ is often referred to as the 'Hallett-Mossop temperature regime'. Heymsfield and Mossop (1984) highlighted the importance of the rimer surface temperature, which can be higher than the air temperature due to the latent heat of crystallization released upon freezing of accreted droplets (Heymsfield and Mossop, 1984). Generally, the freezing of a supercooled water droplet can be subdivided into three stages (Macklin and Payne, 1967). In the initial freezing or recalescence stage, ice dendrites rapidly grow through the droplet starting from the nucleation site, and the latent heat released during the phase transition causes the temperature



of the droplet to rise to the melting point of water (0 °C) (Macklin and Payne, 1967; Pruppacher and Klett, 2010; Korolev
      and Leisner, 2020). In the subsequent second freezing stage, where the remaining liquid water freezes, the droplet temperature
      stays at 0 °C as heat dissipation to the environment via heat conduction balances the latent heat of crystallization. After freezing
      is completed, the droplet cools down to the temperature of the environment. Freezing of a small droplet upon collision with a
      larger ice particle follows the same pathway but with a higher rate (100x) of latent heat removal through the water–ice boundary,

so that the temperature of the droplet may not reach the melting point at all. However, if the droplet mass accretion rate is
      high, the rimer surface temperature could rise to the melting point of water, signifying the transition from dry to wet growth
      regime (Schumann–Ludlam limit, Schumann, 1938; Ludlam, 1958; Pruppacher and Klett, 2010). The wet growth regime is
      thought to inhibit rime-splintering (Bader et al., 1974; Pruppacher and Klett, 2010; Korolev and Leisner, 2020, and references
      therein). Following the initial experiments by Hallett and Mossop, a connection between the droplet size distribution (DSD)

and the rate of SIP due to rime-splintering has been identified in the later experimental studies. In particular, the efficiency of
      rime-splintering was found to be the highest if droplets smaller than 12 $\mu$m and larger than 24 $\mu$m in diameter were present at
      the same time (Hallett and Mossop, 1974; Mossop, 1978, 1985a). The rimer velocity also seems to be a relevant parameter for
      rime-splintering with reported maximum SIP rates observed between 2 m s$^{-1}$ and 6 m s$^{-1}$ (Mossop, 1976, 1985a; Saunders and
      Hosseini, 2001). Despite multiple evidence of the rime-splintering SIP from laboratory experiments and in situ observations,

the mechanism responsible for the release of SI splinters is still debated. Several mechanisms have been proposed that might
      cause SIP during riming based on the release of stresses due to mechanical action (shedding), pressure or thermal gradients
      during riming, freezing initiation of a droplet that makes glancing contact with the rimer or detachment of frail ice needles by
      sublimation (Mossop, 1976; Choularton et al., 1980; Dong and Hallett, 1989).

      Two types of experimental methods have been applied to study rime-splintering in laboratory, either using a single fixed

ice grain simulating realistic graupel (Brownscombe and Hallett, 1967; Aufdermaur and Johnson, 1972; Bader et al., 1974)
      or mimicking falling rimers by large, rotating ice-covered metal rods in a cloud simulation chamber filled with supercooled
      droplets produced by steam from a boiler (Hallett and Mossop, 1974; Mossop, 1976, 1985a; Saunders and Hosseini, 2001).
      In the following discussion, we refer to HM-type experiments corresponding to the latter case. For all of these experimental
      methods, the conditions regarding temperature, DSD and impact velocity were mostly comparable to those encountered in

the atmosphere. However, the information about droplet-rimer collision rates and rimer surface temperature is often missing.
      Simultaneous microscopic observation of the riming surface in an air flow and detection of SI particles have not been per-
      formed to date. In many cases, the existence of background ice particles in the simulation chamber led to large uncertainties
      in determined SIP rates. Difficulties in controlling the experimental conditions have apparently been responsible for the low
      replicability of the measured SIP rates that have been reported to be in the range from 0 to 8000 SI particles per mg rime at

similar conditions (Korolev and Leisner, 2020). It is remarkable that a significant number of SI particles is reported mainly
      from the HM-type experiments using large rimer surfaces and steam generation of cloud droplets, with the exception of the
      study from Latham and Mason (1961). In the latter study, however, the presence of carbon dioxide probably caused the high
      number of splinters (Korolev and Leisner, 2020).



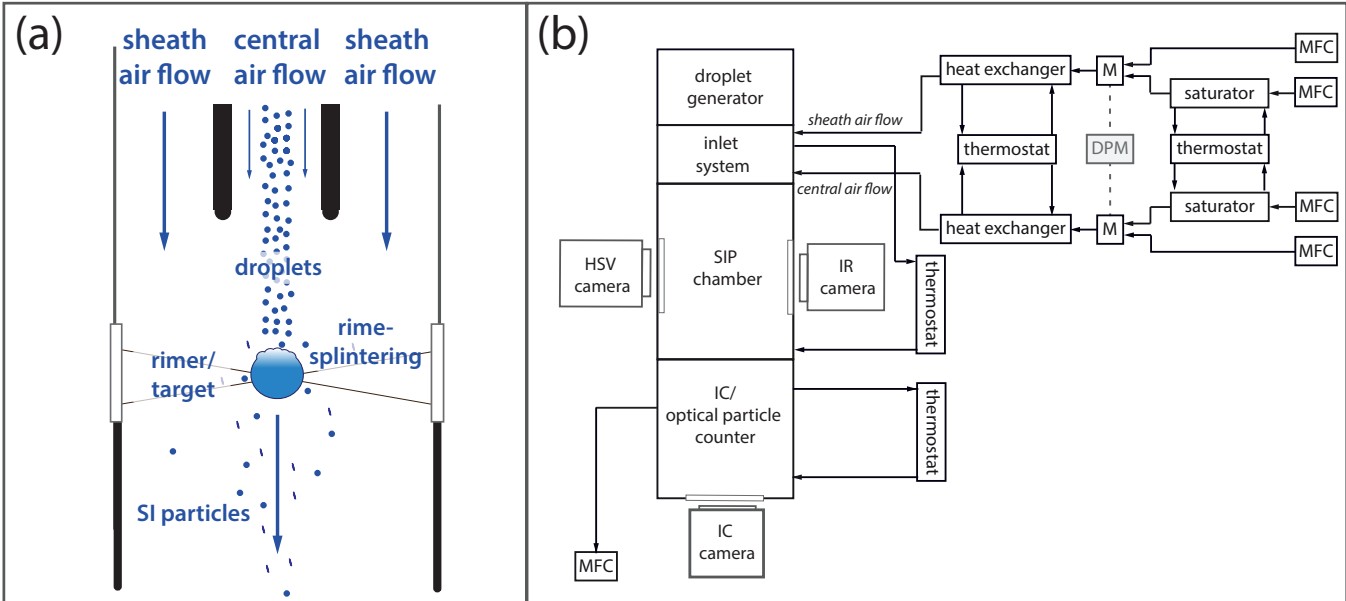

**Figure 1.** Sketches of (a) SIP chamber; (b) schematic of IDEFIX with flow conditioning system, droplet generator, inlet system, SIP chamber with high speed video (HSV) camera, infrared (IR) thermography system and Ice Counter (IC).

The overview of the previous experimental results indicates the necessity to revisit the rime-splintering experiments under better controlled conditions and with improved measuring techniques. For this purpose, the new laboratory experiment IDEFIX (Ice Droplets splintEring on FreezIng eXperiment) was set up, allowing direct observation of the riming process on the surface of a fixed graupel particle with high-speed video microscopy and infrared thermography, while detecting the SI particles with a newly developed ice counter.

## 2 Ice Droplets splintEring on FreezIng eXperiment (IDEFIX)

The experimental setup IDEFIX has been developed to study SIP resulting from riming on a qualitative and quantitative level for atmospheric representative and well-controlled conditions. In IDEFIX, riming is simulated by exposing a fixed large ice particle (diameter $\approx 1\,\mathrm{mm}$) to a stream of supercooled water droplets carried by an air flow. The rimer was produced by freezing a drop of $1\,\mu\mathrm{L}$ deionized water placed at the intersection of two carbon fibers with a thickness of about $6\,\mu\mathrm{m}$ (Fig.1a). The air flow velocity corresponds to the terminal fall velocity of a graupel grain of 1 mm in diameter, which is approximately $1\,\mathrm{m\,s^{-1}}$. IDEFIX provides thermodynamic and flow conditions and allows for visualization of the riming process, measurement of the graupel surface temperature, and quantification of the production rate of the SI particles.



## 2.1 Experimental setup

IDEFIX consists of a pre-conditioning system, a droplet generator, an inlet system, the SIP chamber, and the SI particle detection system (Fig. 1b). IDEFIX is fed with two air flows, i.e., the central and the sheath air flows, each independently conditioned with respect to temperature, humidity and flow rate. Thereto, particle-free dry and humidified air (Nafion saturator, Gasmet) are mixed in turbulent mixing chambers (M) and subsequently cooled to the required temperature inside the heat exchangers controlled by a thermostat (FP50, Julabo). In the experiments described here, the central and the sheath air flows have the same temperature.

The central and the sheath air flows are isokinetically combined in an inlet system upstream of the SIP chamber. Flow conditions inside the SIP chamber are laminar. For a subset of experiments simulating higher droplet impact velocities, the central air flow was increased up to factor of three. The supercooled droplets, generated by a droplet generator, are injected into the central air flow in the inlet system. Upon entering the SIP chamber, the droplets are supercooled for approx. 5 s and in thermal equilibrium with the air flow, as the thermal relaxation time of $20\,\mu$m droplets is on the order of 20 ms. The temperature of the inlet and the metal walls of the SIP chamber are controlled by a thermostat (F25, Julabo). The inner diameter of the SIP chamber (17 mm) was chosen to minimize the potential losses of SI particles to the walls by impaction of splintered SI particles. As worst case scenario we consider a $20\,\mu$m diameter particle ejected from the rimer surface with $10\,\mathrm{m\,s^{-1}}$. In this case the stopping distance is about 7 mm. Downstream the SIP chamber either an optical particle counter WELAS® (WELAS®1000, PALAS®), or the newly developed ice particle counter, is installed to determine the DSD, or the total number of SI particles, respectively.

In this study, the air flow temperature was varied in the range between $-4\,°$C and $-10\,°$C and the air flow velocity between $1\,\mathrm{m\,s^{-1}}$ and $3\,\mathrm{m\,s^{-1}}$. Due to the systematic temperature deviation at the rimer position of around +1 K, the water vapor saturation with respect to ice is 90%. Water vapor emitted by evaporating water droplets contributes less than 1% RH and can be neglected.

## 2.2 Droplet generation and size distributions

A monodisperse droplet generator (MDG, model 1530, TSI®) is used for generating droplets with different size distributions. In the MDG, mechanical vibration of the nozzle combined with aerodynamic focusing produces a jet of deionized water droplets (details are described in Duan et al., 2016). The DSD and total droplet number can be controlled by adjusting the liquid flow rate, vibration frequency, and the focusing air flow rate. To study the impact of different droplet sizes on the efficiency of rime-splintering SIP, four different MDG settings were used and the resulting DSDs were measured with an optical particle counter WELAS®. The MDG settings and parameters of the log-normal fits of the DSDs are given in Table 1 and shown in Fig. 2.

As the droplets produced by the MDG are intrinsically charged, two bipolar corona discharges operating at 50 Hz alternating voltage of 5 kV were used to partly neutralize the droplets prior to entering the inlet system. This has significantly improved the stability of droplet generation.





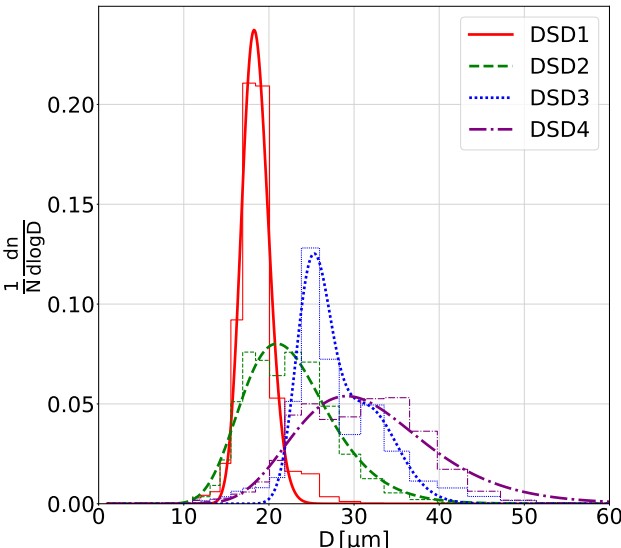

**Figure 2.** Normalized number size distributions of four different droplet populations used in the IDEFIX experiments (DSD1–DSD4) including measurements (bar graph) and respective log-normal fits (curves, parameters are given in Table 1).

**Table 1.** MDG settings frequency F, liquid flow rate Q and flow focusing pressure difference ff resulting in four different droplet size distributions DSD1–4, which were found to be log-normally distributed. Given are the geometric mean diameter $D_g$ and geometric standard deviation $\sigma_g$ of the log-normal distributions fitted to DSD1–4 (Fig. 2). For DSD3 a bimodal fit was applied. The goodness of fit is described by the coefficient of determination $R^2$.

|  | MDG settings | | | Fit parameter | | |
|---|---|---|---|---|---|---|
| DSD | F | Q | ff | $D_g$ | $\sigma_g$ | $R^2$ |
|  | [kHz] | [mL h$^{-1}$] | [psi] | [$\mu$m] | [$\mu$m] |  |
| DSD1 | 220 | 2 | 2.6–2.8 | 18.4 | 1.6 | 0.99 |
| DSD2 | off | 2 | 2.6–2.8 | 22.1 | 5.5 | 0.97 |
| DSD3 | 100 | 2 | 1.4–1.6 |  |  |  |
| mode1 |  |  |  | 25.3 | 1.9 | 0.97 |
| mode2 |  |  |  | 32.1 | 3.1 | 0.97 |
| DSD4 | off | 2 | 1.4–1.6 | 31.0 | 8.4 | 0.94 |

## 2.3 Riming observation

For microscopic and thermal observation of the riming process a high speed video (HSV) camera (Phantom Veo 710L, HSVision) and an infrared camera (IR, ImageIR 7340, InfraTec GmbH) were used. The HSV camera was operated with a 10x microscopic objective (Plan Apo, Mitutoyo) in the transmitted illumination. This setup allowed for an exposure time of $2\,\mu$s,

130





a focal depth of about 200 μm and a pixel resolution of about 2 μm. The field of view (FOV), as well as the recording time,
135   varied according to the selected frame rate. For low frame rates in the range of 100–1000 fps the maximum frame size was
1280 x 800 Pixel corresponding to the FOV of 3.5 x 1.6 mm. With this setting, it was possible to observe the evolving rimer
surface structure over a time period of several seconds. In order to record individual droplet–rimer collisions with high frame
rates (up to 70000 fps), the maximum frame size was reduced to 256 x 256 Pixel (FOV 0.5 x 0.5 mm).

    The IR camera was operated with a 2x germanium macroscopic lens and provided measurements of the rimer temperature
with accuracy of ±1 K, based on the factory calibration in the temperature range from −30 °C to 300 °C, as described in
Kleinheins et al. (2021). In the IR calibration experiment (Appendix C), the rimer surface of true temperature 0 °C was found
to appear 1.4±0.6 K colder in the IR measurements. This offset, which can be attributed to the presence of Ge-windows and
proximity of chamber walls, should be kept in mind when considering the diagrams showing the IR temperature values. The
IR video sequences have been recorded with a frame rate of 25 fps.

**2.4   Ice particle detection**

    To count the SI particles, a custom-build Ice Counter (IC) is installed downstream of the SIP chamber of IDEFIX. The cross
section of the IC is shown in Fig. 3a. SI particles and droplets carried by the air flow are directed onto the surface of a
sucrose solution (Merck™, 42.85 wt %) kept at a temperature just below its melting point. As the sucrose solution is slightly
supercooled, impinging ice crystals grow slowly to optically detectable sizes whereas liquid droplets dissolve in the solution
upon contact. The melting point for this sucrose concentration was determined experimentally to be −5.0 °C and the solution
was kept at −5.7 °C throughout the experiments. The method of using a supercooled sugar solution for ice crystal detection
was first introduced by Bigg (1957) and has been popular for detecting ice crystals in laboratory experiments (e.g. Mason and
Maybank, 1960; Aufdermaur and Johnson, 1972; Kolomeychuk et al., 1975). By heating the sugar solution above the melting
point, ice crystals melt and a new experimental cycle can be started without the need to exchange the sucrose solution. The
sucrose bath is illuminated from above with two white LEDs and the ice crystals floating on the surface of the sucrose solution
are observed with a video camera (Photonfocus MV1-D2080-160-G2) through the transparent windows of the cooling cell
from below. An example of ice crystals grown in the IC bath is shown in Fig. 3b. The impaction probability of an ice crystal
on the surface of the sucrose solution is higher for larger ice crystals and higher flow velocity. The flow in the IC is accelerated
through a nozzle of 3.5 mm diameter at the end of the conical flow tube connected to the SIP chamber (see Fig. 3a). At this
nozzle size and a total air flow rate of 12.35 L min$^{-1}$ the impaction cut-off aerodynamic diameter of 2 μm has been found
experimentally. However, taking into account that secondary ice crystals are exposed to subsaturated conditions with respect
to ice and sublimate on their way to the IC, the 2 μm cut-off diameter at the collision point would correspond to an initial ice
particle diameter of 3.3 μm released in the vicinity of the rimer. Details of the IC characterization experiments can be found in
Sect. A1 in the Appendix. To ensure that there are no background counts, every experiment has been preceded and followed
by a blank test, where a rimer was situated in the SIP chamber and exposed to a droplet-free air flow for at least about 2 min.
Only the experiments where no ice appeared in the IC sucrose solution are considered as valid.



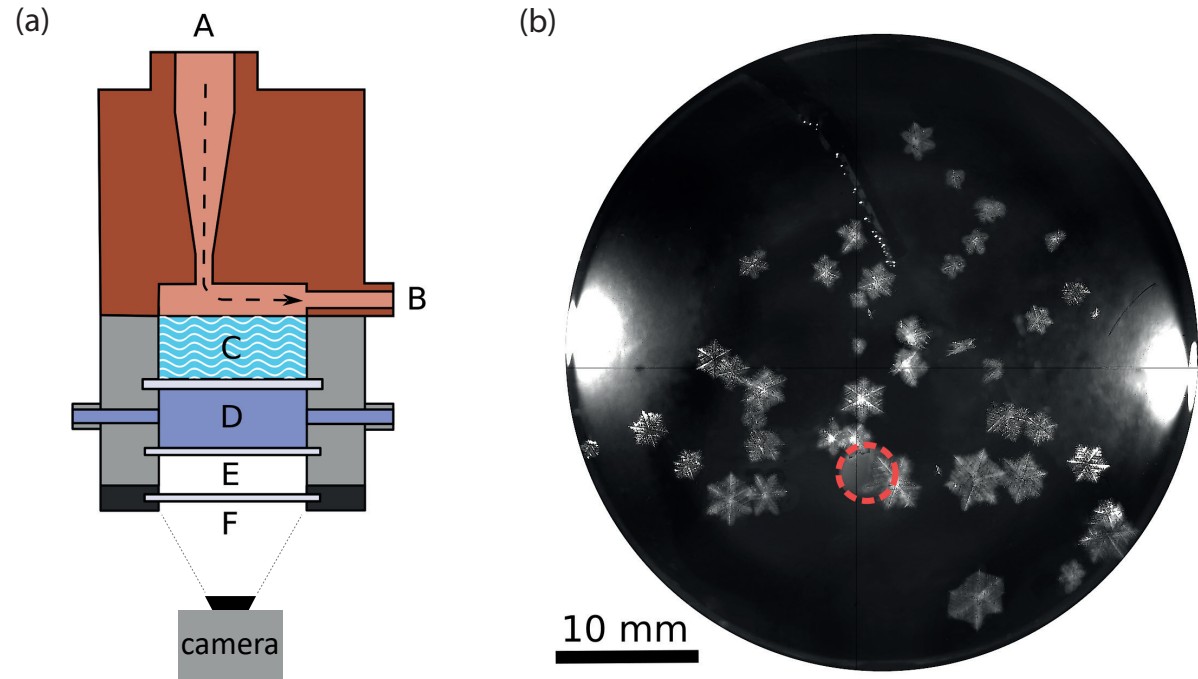

**Figure 3.** Ice Counter (IC) used to detect the SI particles via impaction on supercooled sucrose solution. In panel (a), a cross-section of the IC is shown including inlet (A), outlet (B), sucrose bath (C), flow-through chamber with cooling liquid (D), thermal isolation cell with two transparent windows (E) and camera setup (F). The entire IC housing is cooled by ethanol flowing through the chamber (D) and the copper housing body (not shown). In panel (b), an example of ice crystals in the supercooled sucrose solution is given. The bright spots on the left and right side are the LED lights installed in (C) for illumination. A PT-100 temperature sensor is immersed into the sucrose solution to measure its temperature. The nozzle position is marked by the dashed red circle.

## 3 Results and discussion

### 3.1 Riming

**Growth Regimes**

Riming experiments were conducted with four different droplet populations as described in the experimental section. For each DSD, the respective collision rates have been calculated as described in the Appendix (see Sect. B1+B2) and can be found in Table 2 together with the relevant experimental parameters. Depending on the settings (DSD1–4, see Fig. 2), three different riming regimes were observed at IDEFIX: dry (a), wet (b) and transition growth (c). Examples of the characteristic time series of the rimer surface temperature before, during and after riming are given in Fig 4.

Dry growth was observed for low mass collision rates associated with DSD1 and DSD2. In all experiments with these settings, the rimer surface temperature increased by a constant value of about 0.3 K to 2 K during riming, as shown in Fig. 4a.



**Table 2.** IDEFIX parameter space for SIP experiments.

| | |
|---|---|
| Air flow temperature | $-4\,°C$, $-5\,°C$, $-7\,°C$, $-10\,°C$ |
| Maximum air flow velocity | $1\,m\,s^{-1}$, $3\,m\,s^{-1}$ |
| Ice particle (rimer) diameter | $\approx 1\,mm$ |
| Range of droplet diameters | $10$–$50\,\mu m$ |
| Avg. number collision rate | $(3.6$–$8.7) \times 10^2\,mm^{-2}\,s^{-1}$ |
| Avg. mass collision rate | $(1.6$–$16) \times 10^{-3}\,mg\,mm^{-2}\,s^{-1}$ |
| Impaction cut-off diam. in IC | $2.0\,\mu m$ |
| Ice crystal detection limit | $3.3\,\mu m$ |

After riming, the rimer surface temperature returns to the environment temperature. In the wet growth regime (Schumann, 1938) resulting from high mass collision rates, the latent heat released by freezing of continuously colliding droplets cannot be removed rapidly enough by heat dissipation and sublimation, so that the surface temperature of the rimer rises to the melting
point of water (see heat balance model of a riming particle in Appendix Sect. B1) and a liquid layer forms at the graupel surface (Fig. 4b). In the cases where the collision rate varied with time, the rimer surface temperature was observed to oscillate between the melting point and a lower temperature. We describe this as transitional growth regime (see Fig. 4c). Wet growth as well as the transitional growth regime was occasionally observed with DSD3 and DSD4.

**Microscopic structure of rime**

In the dry growth regime, the structure of rime is independent of the collision rate, because the individual droplets colliding with the rimer freeze completely before the next droplet arrives. In the experiments described here, the inter-arrival time between two consecutive droplets hitting the same spot on the rimer surface was between 0.4 and 0.6 s, as calculated with Eq. D3 in Appendix Sect. D. This is significantly longer then the time of individual droplet freezing which was calculated to be between 0.01 s and 0.04 s, (Eq. D1,D2 in Sect. D) assuming droplets with semi-spherical cap geometry freezing on ice surface at a
temperature between $-10\,°C$ to $-5\,°C$, respectively.

Upon collision with the rimer surface, a droplet starts spreading to assume the equilibrium shape determined by the surface energy relationship between ice, water, and gas phase. However, the spreading is counteracted by freezing, which is also initiated at the moment of contact. Contrary to the freezing rate, the spreading rate does not depend on the temperature in the range between $-3\,°C$ and $-8\,°C$; the final shape of an accreted droplet is therefore defined by the time required to halt the
spreading. The HSV records taken at $-5\,°C$, $-7\,°C$ and $-10\,°C$ in Fig. 5 show different shapes of accreted droplets. At $-10°C$, the accreted frozen droplets retain their quasi-hemispherical shape with apparent contact angle between $80°$and $120°$. Multiple frozen droplets are forming narrow ice spicules at this temperature. At temperatures above -10 °C, droplets spread to flat lentil shapes with apparent contact angles of about $30°\pm10°$, forming thick rime columns.





**Figure 4.** Sequence of IR images A) before, B) during and C) after riming (upper panel) and time series of the average rimer surface temperature measured within circles of 150–200 $\mu$m diameter close to the rimer top in the IR images for dry (a), wet (b) and transitional growth regime (c). In example (c), several measurement circles (C1–C7, numbered from bottom to top) were applied. The red shaded areas mark the IR temperature region of the water melting point according to calibration measurements. Please note the different scales of the color bars.

The tendency to freeze in spherical shapes upon droplet accretion at lower temperatures was also observed by Macklin and
Payne (1968, 1969) and Dong and Hallett (1989). The microscopic photographs of rime structures by Griggs and Choularton (1986) also show preferential formation of fragile ice needles with decreasing temperature.

During riming in the transitional or wet growth regime, a dense ice particle with a smooth surface was formed at all investigated temperatures (Fig.6, lower panel). In these growth regimes, the characteristic inter-arrival time is about 0.10 s and is





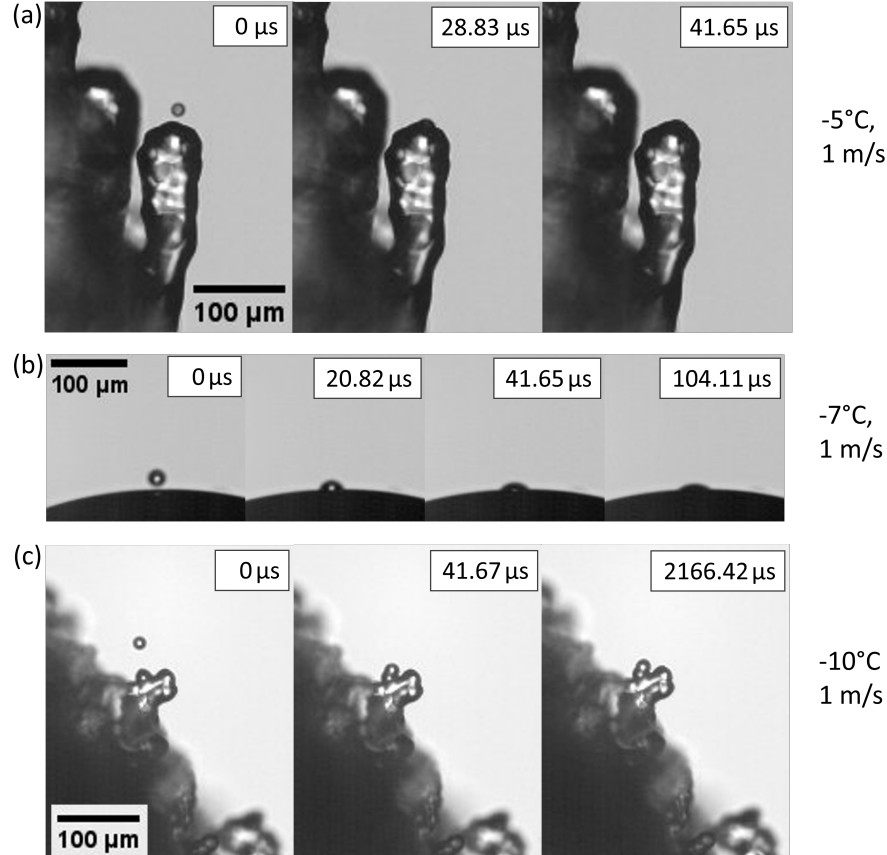

**Figure 5.** HSV-image sequence of individual droplet–rimer collisions with air flow velocity $1\,\mathrm{m\,s^{-1}}$ at $-5\,^{\circ}\mathrm{C}$ and droplet diameter $D = 16.5\,\mu\mathrm{m}$ (a), $-7\,^{\circ}\mathrm{C}$ and $D = 20\,\mu\mathrm{m}$ (b) and $-10\,^{\circ}\mathrm{C}$ and $D = 18.5\,\mu\mathrm{m}$ (c) in the dry growth regime.

close to the freezing time of water droplets in the size range of $30\text{–}50\,\mu\mathrm{m}$ in diameter for the investigated temperature range in the IDEFIX experiments (approx. 0.08–0.23 s at -5 °C). Hence, larger accreted droplets are still partially liquid when the next droplets arrives at the same spot. This leads to continuous wetting of the rimer surface, thus impeding the growth of rime spires.

## 3.2 Rime-splintering

Out of 30 valid IDEFIX experiments with droplet–rimer mass collision rates in the range between $1.6 \times 10^{-3}\,\mathrm{mg\,mm^{-2}\,s^{-1}}$ to $16 \times 10^{-3}\,\mathrm{mg\,mm^{-2}\,s^{-1}}$, only in six experiments potential SIP events were identified by observing ice particles in the IC (cf. Table 3). A detailed overview of the valid rime-splintering experiments as well as HSV and IC images of the experiments with potential SIP are given in Table E1 and Fig. E1 in the Appendix. In two experiments at $-5\,^{\circ}\mathrm{C}$, ice particles were counted with the IC during riming, whereby no co-occurring SIP were seen in the high-speed video recordings. There were also four



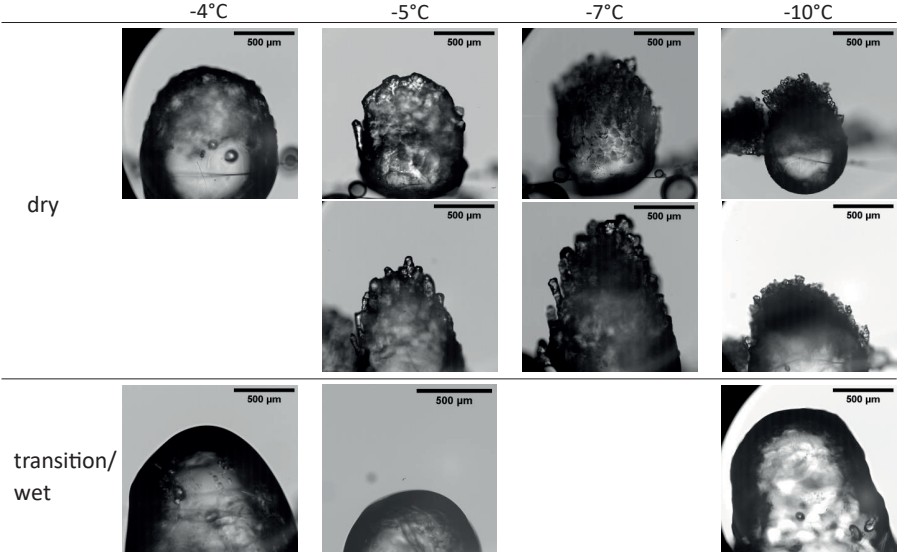

**Figure 6.** Macroscopic view on the graupel surface structure in a matured stage grouped by temperature and prevailing growth regime. In all cases the air flow velocity was set to $1\,\mathrm{m\,s^{-1}}$.

**Table 3.** Number of valid IDEFIX experiments in total and where ice crystals were detected in the IC (potential SIP) during or after riming for all experimental temperatures. The number of cases in the different growth regimes is given in parentheses. In case of potential SIP, the number of detected ice crystals and the prevailing growth regime during the experiment is given as well. Further details can be found in Table E1.

| | | $-4\,°\mathrm{C}$ | $-5\,°\mathrm{C}$ | $-7\,°\mathrm{C}$ | $-10\,°\mathrm{C}$ |
|---|---|---|---|---|---|
| **total** | | **4** | **11** | **6** | **9** |
| (dry/wet/transition) | | (3/0/1) | (9/1/1) | (6/0/0) | (6/0/3) |
| | **during riming** | **0** | **2** | **0** | **0** |
| | observed number of ice crystals | | 20 (dry) 5 (dry/transition) | | |
| potential SIP | **after riming** | **0** | **0** | **1** | **3** |
| | observed number of ice crystals | | | 1 (dry) | 1 (dry) 2 (dry) 1 (transition) |

experiments at $-7\,°\mathrm{C}$ and $-10\,°\mathrm{C}$, in which one or two individual ice crystals were detected in the IC several minutes after
riming. Among these four cases, sublimation induced break-off of a rime spire was observed twice with the HSV camera. An





example of a rime-spire bending down before its final break-off, which was documented after about 10 min of sublimation, is shown in Fig. 7.

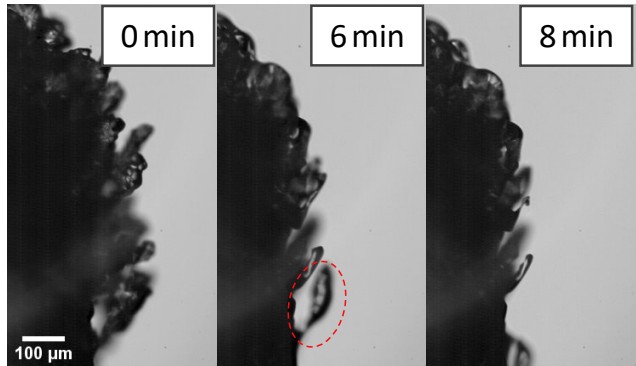

**Figure 7.** High speed image sequence of sublimating rime spires on the surface of a graupel particle at $-10\,°\mathrm{C}$ in an ice subsaturated environment with air flow velocity $1\,\mathrm{m\,s^{-1}}$ after riming. After about 8 min of sublimation, the encircled rime-spire was observed to bend down, before it finally broke off two further minutes later.

The cases for which ice crystals were observed in the IC during riming could not be reproduced in other experiments conducted under the same conditions. We therefore conclude, that no efficient and reproducible SIP was observed during riming experiments with IDEFIX within the investigated parameter range (see Table 2). In all potential SIP cases, the number of ice crystals detected in the IC was well below the values expected on the basis of the original HM experiments (Hallett and Mossop, 1974; Mossop, 1976, 1985a) in the temperature range of $-3\,°\mathrm{C}$ to $-8\,°\mathrm{C}$. In their experiments, up to 300–350 SI particles per mg rime were observed at around $-5\,°\mathrm{C}$ for rimer velocities in the range of $2$–$4\,\mathrm{m\,s^{-1}}$ and accretion rates on the order of $10^{-4}\,\mathrm{mg\,mm^{-2}\,s^{-1}}$ (Mossop, 1985a). Summarizing all SIP experiments in the dry growth regime at -5 °C, the number of detected ice crystals account for maximum 7.6 SI particles per mg. Note, that this derived SIP rate is determined by one out of five experiments (Table 3). We acknowledge the possibility that all SI crystals generated via rime-splintering were considerably smaller than $3.3\,\mu\mathrm{m}$ diameter and therefore could not be detected. The possible reasons for inefficient SIP in our experiments are summarized at the end of this section. In the following, we discuss the individual mechanisms potentially underlying the rime-splintering SIP (illustrated in Fig. 8) based upon our microscopic observation of riming events in more detail.

*Splintering during freezing of an accreted droplet.* This mechanism suggests that the SI particles are produced as a result of the fragmentation of ice shell forming around a freezing droplet (Mossop, 1976; Choularton et al., 1980; Griggs and Choularton, 1983) in analogy to shattering of droplets freezing in free fall (Kleinheins et al., 2021; Lauber et al., 2018; Keinert et al., 2020). As an ice shell forms around the freezing droplet, the pressure in the liquid water trapped inside increases considerably (up to 240 bar, Kleinheins et al., 2021). If pressure-induced stress exceeds tensile strength of ice, pressure is released by elastic deformation followed by fragmentation of the ice shell. An illustrative case of pressure induced fragmentation and splintering





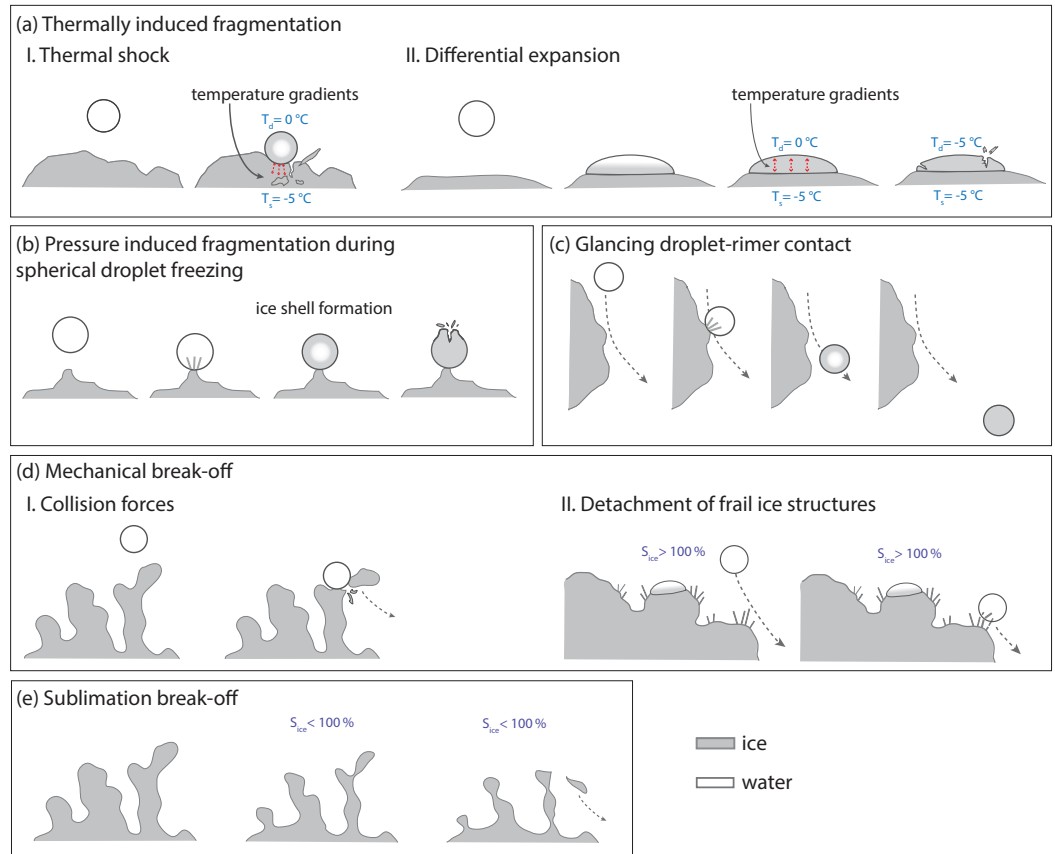

**Figure 8.** Schematic illustration of different mechanisms proposed to explain the rime-splintering SIP mechanism (a–d). A proposed mechanism, where riming leads indirectly to SIP is sketched in (e). In (a), temperature gradients are indicated with red arrows. The ice surface has an exemplary temperature of $T_s = -5\,°C$ just like the incident supercooled droplet. During freezing upon impact, the droplet temperature ($T_d$) increases up to $0\,°C$. Schematics (a-II) is adapted from the simplistic conceptual model and the observation of a shattered droplet from Dong and Hallett (1989, Fig. 12, 13). (b) is adopted from a camera observation of splintering due to ice shell fragmentation of a spherically shaped accreted presented in Choularton et al. (1980, Fig. 2c). In (d-II), saturation with respect to ice ($S_{ice}$) has to be over $100\,\%$ to enable depositional growth of frail ice structures on the rimer surface. In an ice subsaturated environment (e), sublimation can lead to the fragmentation of rime spires previously grown on the graupel surface.

of a droplet accreted on a surface covered with ice at $-7\,°C$ was observed by Choularton et al. (1980, Fig. 2c therein) and is shown schematically in Fig. 8b.

Formation of an ice shell enclosing a freezing droplet requires spherically symmetrical removal of latent heat of crystalliza-
tion through the droplet surface, facilitated e.g. by ventilation and droplet rotation in free fall. A stationary droplet accreted on the surface of a rimer could form an ice shell only if the heat flux through contact surface is comparable with that of diffusion and convective heat removal through the air. This could be the case if the freezing droplet is connected to the rimer by a thin



neck, formed, for example, by a smaller frozen droplet present at the point of contact (Choularton et al., 1980; Emersic and Connolly, 2017).

Observations obtained with IDEFIX and reported in previous studies of Dong and Hallett (1989) and Emersic and Connolly (2017) have shown that the droplets tend to spread upon impact on smooth and rough ice surfaces at temperature above $-10\,°\mathrm{C}$. This clearly contradicts the hypothesis of the pressure-induced fragmentation of the ice shell forming around the freezing droplet accreted to the rimer surface. Below $-7\,°\mathrm{C}$, there is a distinct tendency of the accreted droplets to freeze in a spherical shape. However, in our experiments with IDEFIX, no SIP during riming was observed at $-10\,°\mathrm{C}$. To this effect,
Griggs and Choularton (1983) speculated that at such low temperatures, the ice shell of an accreted freezing droplet might be too strong for cracking. This, contradicts the observation of Kleinheins et al. (2021) who reported pressure release events in freezing droplets down to the temperature of $-25\,°\mathrm{C}$ for much larger droplets.

Even if the ice shell is not forming around the freezing droplet, the latent heat released during freezing can induce thermal gradients at the droplet–ice interface and thus lead to differential thermal expansion of ice, which could result in fragmentation
of the freezing accreted droplet or the underlying ice structure, as illustrated in Fig. 8a-I and Fig. 8a-II, respectively. A detailed review of these thermal shock mechanisms is given in Korolev and Leisner (2020) and references therein.

Although the dry growth regime would have offered suitable conditions, no evidence was found for such SIP mechanisms associated with thermal shock (Koenig, 1963; King and Fletcher, 1976a) or shear stress release (Dong and Hallett, 1989) in IDEFIX experiments. Even if the large fraction of SI particles generated in this way would be smaller than $3.3\,\mu\mathrm{m}$ in diameter
and have detection probabilities less than $50\,\%$ considering sublimation effects on the way to the IC, we expect that due to the high number of droplet collisions at IDEFIX (on the order of 2000 per experiment), at least a few SI particles would have been detected with the IC.

*Droplet freezing induced by glancing contact with rimer surface*, illustrated in Fig. 8c, can be considered as a SIP mechanism (Mossop, 1976). According to our statistical estimation, about 2 to 17 droplets per second are passing near the rimer
surface at a distance smaller than the median droplet diameter of $20$–$30\,\mu\mathrm{m}$ (concept described in Wang, 2013). Based on this estimation, multiple frozen droplets should have been observed in the IC as their size would ensure their efficient impaction on the surface of the sucrose solution. As we do not observe continuous SIP, droplets experiencing a glancing contact with the rimer surface are always accreted by the surface or do not freeze upon glancing contact with ice. On a rare occasion when a droplet on a glancing trajectory was in the focal plane of the imaging optics, we have observed droplet coalescence with the
rimer surface (Fig. 9). The preferential accretion of grazing droplets is affirmed by the results of Emersic and Connolly (2017).

Another potential SIP associated with riming is *break-off of frail ice structures due to mechanical action or sublimation*. Fragile ice formations such as chains of frozen droplets or towers on the rimer surface (also called rime spires) or ice needles preferentially growing by vapor deposition on the rimer surface at around $-5\,°\mathrm{C}$ in an ice and water supersaturated environment (Libbrecht, 2017) are suggested to be a further potential source of SI particles associated with rime-splintering SIP (Macklin,
1960; Bader et al., 1974; Mossop, 1976). Those fragile ice structures may break off upon collisions with droplets or other ice particles (Mossop et al., 1974), as illustrated in Fig. 8d-I,II. Neither detachment of frail ice needles, nor mechanical break-off of rime spires was observed in IDEFIX in the investigated parameter space. Even at $-10\,°\mathrm{C}$, the more fragile rime spires are



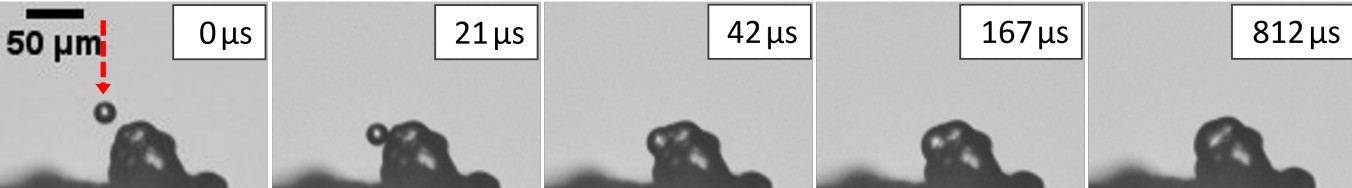

**Figure 9.** Glancing collision of a $20\,\mu$m in diameter droplet resulting in accretion to the rimer surface at $-7\,°C$ and $1\,m\,s^{-1}$. The arrow illustrates the previous trajectory of the droplet.

mechanically stable. Experiments by Griggs and Choularton (1986) have shown that rime spires are unlikely to break off due to sheer forces from air motion alone, and that relative velocities of above $60\,m\,s^{-1}$ are required for this to occur. Deposition
growth of ice needles could not be observed at IDEFIX as the flow around the rimer is slightly subsaturated with respect to ice. However, very few cases of rime spire break-off due to sublimation were observed in IDEFIX after riming at $-7\,°C$ and $-10\,°C$ (cf. Table 3).

Sublimation break-off of rime spires can take place in ice subsaturated conditions and thus might be a mechanism for SIP (Mossop and Hallett, 1974; Oraltay and Hallett, 1989). Thereby, thinner parts of a rime spire or other fragile ice structure
sublimate faster compared to thicker ice structures leading to ice particle separation and consequently to ice multiplication (Fig.8e). Sublimational break-off was described for pristine ice crystals with aspect ratios larger than 3 favorable and rimed ice particles (Dong and Hallett, 1989; Dong et al., 1994; Bacon et al., 1998; Korolev and Leisner, 2020).

Generally, fragmentation by sublimation can be thought as separate SIP process from rime-splintering (Korolev and Leisner, 2020), since riming plays only an indirect role leading to a finely structured graupel surface dominated by frail rime spires in
the dry growth regime. Based on the argument of Korolev and Leisner (2020) and Korolev et al. (2020) that small ice fragments in a subsaturated cloud environment are more likely to fully sublimate before they return to a cloud zone supersaturated with respect to ice, it might be unlikely that this mechanism is important in atmospheric clouds. In contrast, Deshmukh et al. (2022) could theoretically derive a significant contribution of SIP due to sublimation taken also graupel into account. It is conceivable that the mechanism may be important in regions near the cloud edge where entrainment of dry air occurs, as discussed earlier
by Bacon et al. (1998).

To discuss the possible reasons why no efficient SIP has been observed in the IDEFIX experiments in contrast to earlier HM-type experiments (Mossop et al., 1974; Mossop, 1976, 1978, 1985a; Saunders and Hosseini, 2001), we compare the details of experimental set-ups. In the IDEFIX riming experiments, (i) the geometrical cross-sectional area of the rimer is
about 1000 times smaller than in the HM-type setup, (ii) the droplet populations contained almost no droplets smaller than $12\,\mu$m in diameter, (iii) the mass collision rates are a factor of 10 to 100 higher than in the original HM experiments, and (iv) the air flow in IDEFIX is not supersaturated with respect to ice. Nevertheless, the IDEFIX setup reproduces most of the processes that were previously thought to be responsible for SIP via riming splintering.





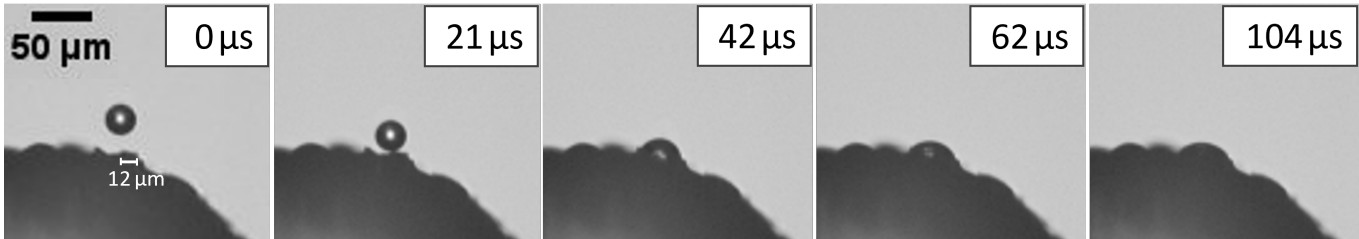

**Figure 10.** Counterexample to spherical droplet freezing on small ice structure. Collision of a 25 $\mu$m in diameter droplet with a 12 $\mu$m wide ice neck at $-7\,°C$ and $1\,\mathrm{m\,s^{-1}}$.

Whereas (i) in IDEFIX the rimer has a realistic graupel size of roughly 1 mm diameter, the rimer was unrealistically large in
the previous HM-type experiments, where cylindrical metal rods (e.g. 30 cm x 0.24 cm) were used as rimer (Hallett and Mossop, 1974; Mossop, 1976, 1985a; Saunders and Hosseini, 2001). Such a large riming surface would allow observation of effects that are statistically less frequent, concealing the actual mechanism behind the HM process: whether SIP is a continuous process producing low numbers of SI particles over the whole period of riming or a random burst event producing a high number of SI particles. In the second case, a rime-splintering event could be rare but produce a big number of SI particles, for example
if a rimer has to undergo multiple sublimation-deposition growth cycles to develop frail dendrites that can easily break off. To detect such events, significantly higher accretion rates, longer observation times, or a larger number of rimers need to be investigated. For the IDEFIX conditions, the accretion rate cannot be increased without undergoing dry–wet growth transition. Longer observation times at the IDEFIX riming rates or multiple rimers are not feasible with IDEFIX. Consequently, to be able to detect rare but efficient events a different experimental set-up would be required.

(ii) Only the IDEFIX setting with DSD2 contained droplets smaller than 12 $\mu$m. In that case, the concentration ratio of small accreted droplets compared to droplets larger than 24 $\mu$m was 0.05. This is lower than the concentration ratios from 0.1 to 2.0 (or higher) used in HM-experiments, in which the efficiency of rime-splintering was found to depend on the accretion rate of droplets smaller than 12 $\mu$m and larger than 24 $\mu$m in diameter. The described correlation supported the hypothesis that spherical freezing might occur when a large droplet is accreted onto an already frozen small accreted droplet, leading to
spherical freezing and ice shell fragmentation (Griggs and Choularton, 1983; Mossop, 1978, 1985a). Although the influence of small droplets on rime-splintering can not be excluded, our observations indicate that a larger droplet accreting on a smaller ice structure on the rimer surface would spread instead of freezing as a spherical droplet. This is demonstrated by the case displayed in Fig. 10, where a droplet of 25 $\mu$m diameter spreads over a narrow elevated ice structure with a characteristic length of 12 $\mu$m at $-7\,°C$, rather than freezing spherically. Similar observations have been reported by Emersic and Connolly
(2017) (Fig. 7a,b therein), where the spreading of two droplets with diameters of 30 $\mu$m on a frozen droplet cap of about 11 $\mu$m in diameter was observed at rimer temperatures of $-7\,°C$ to $-8\,°C$.

(iii) The IDEFIX mass collision rates in dry growth regime are a factor 10 higher compared to those reported by Mossop (1985a). However, the characteristic time of individual droplet freezing is always much shorter than the characteristic inter-arrival time of colliding droplets (see Sect. 3.1). Thus, in the dry growth regime the rimer surface is completely frozen and



in thermal equilibrium between the two consequent droplets colliding at the same site. From this point of view, the actual
collision rate is not affecting the SIP efficiency as long as the growth regime remains dry. The accretion rate is only relevant
for statistical quantification of the number of SI particles produced per mg rime.

(iv) In contrast to the HM-type experiments where droplets were produced by a steam generator, the humidified air flow in
IDEFIX was slightly subsaturated with respect to ice. Therefore, no depositional growth of ice on the rimer surface could be
observed. It should be noted, however, that for the frail ice structures (dendrites, needles, columns or prisms) to grow to the size
where they could be detached upon collision with a droplet (Fig. 8d,II), a significant time is required. In the middle of the HM
SIP temperature interval ($-5\,°C$) and at water saturation, an ice needle needs about $5\,s$ to reach a length of $10\,\mu m$. During this
time, the growing ice crystal would experience on average more than 10 collision events with a liquid droplet under IDEFIX
conditions (see discussion in the section 3.1) and between 0 and 2 collisions in a former HM-type experiment (considering
accretion rates given in Mossop, 1985a). Thus, an ice crystal growing via deposition of water vapor at water saturation has no
chance to reach the size where it could be mechanically detached under IDEFIX experimental conditions, and only a slight
chance under conditions present in the past HM-type experiments. Therefore, it remains an open question whether detachment
of frail ice structures growing on the rimer surface via water vapor deposition could be an explanation for the high number
of SI particles observed in the previous HM-type experiments as suggested by Mossop (1976). Note, however, that in a real
atmospheric cloud, firstly, depositional ice growth is faster due to the lower gas pressure, and secondly, a falling graupel could
experience strong variations of the accretion rate so that the frail ice structures might have time to develop.

We therefore conclude, that in spite of the difference between the experimental conditions of HM-type experiments and
IDEFIX, the majority of the mechanisms (see points i to iii) supposedly underlying the effective SIP are not supported by our
observations. The role of frail ice structures growing on the rimer surface via water vapor deposition remains an open question.
Previous HM-type experiments were limited to riming in the dry growth regime, because the formation of liquid layer dur-
ing wet growth is thought to inhibit suitable conditions for rime-splintering (Pruppacher and Klett, 2010; Korolev and Leisner,
2020). Interestingly, at -5 °C one of the equivocal cases indicating a potential SIP mechanism was observed during riming in
the transient growth regime. Occasionally, small ice spicules (about $20\,\mu m$ in length) were growing out of the freezing rimer
surface after transition from wet to dry growth regime caused by fluctuation of collision rates at $-10\,°C$. Such ice spicule
growth is illustrated in Fig. 11a–c,e. We hypothesize, that liquid water becomes entrapped in the pockets under the ice shell on
the rimer surface during a change in growth regime from (local) wet to dry, causing internal pressure build-up and the spicule
formation. Although no ice particles were detected in the IC in any of these cases, ice spicule formation could be a source of
SI particles in analogy to SIP during freezing of large droplets. A similar case of spicule formation during wet growth has been
described before only once in Macklin (1960). We have also observed gas bubbles appearing on the surface of ice shell of the
frozen rimer after transition from wet to dry growth regime (Fig. 11d), indicating that the transient growth regime could be
more important for rime-splintering SIP than previously recognized.

Rotation is required to form natural appearing graupel. In the IDEFIX experimental setup, the rimer is fixed by two crossing
carbon fibers, excluding the random movement and precession that a natural graupel particle experiences during free fall.





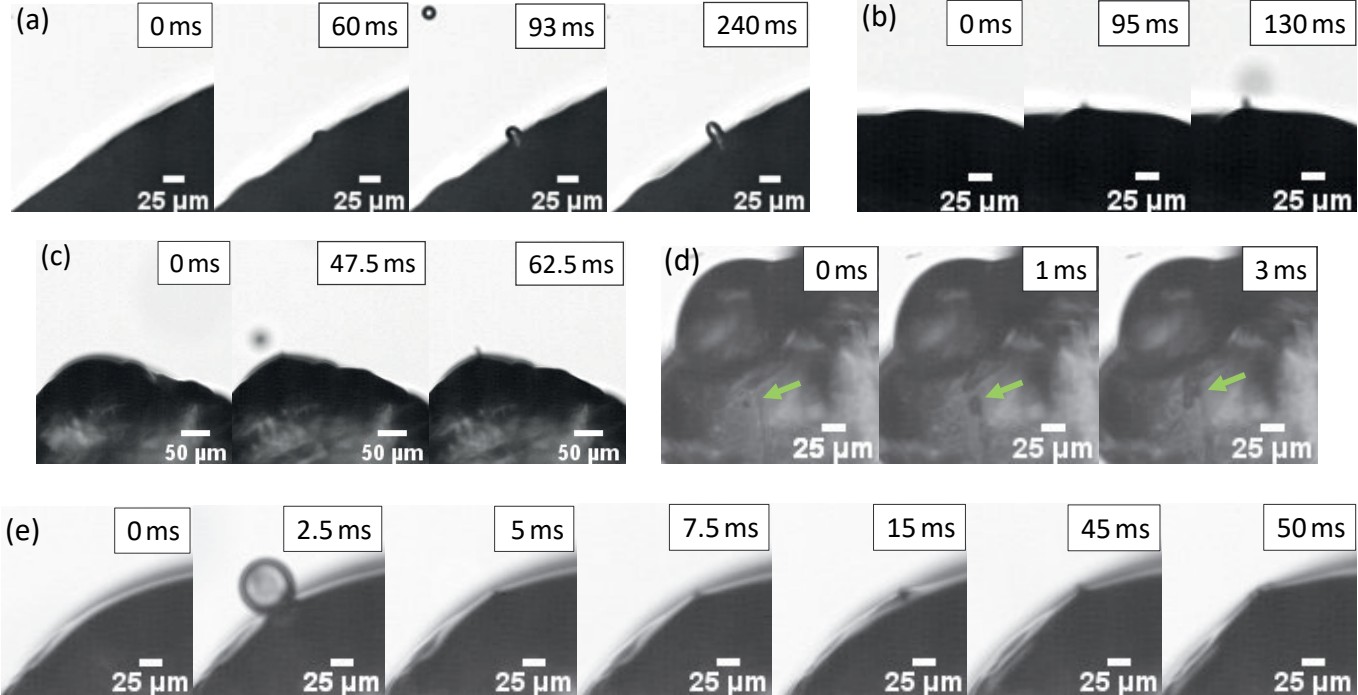

**Figure 11.** IDEFIX observations of ice spicule formation during transitional growth (a–c,e) and of ascending air bubbles within a liquid channel inside an ice target (d) at $-10\,°C$.

According to the time scale considerations given above, local microphysical processes on the rimer surface should not depend on the collision rate in the dry growth regime. Moreover, a free-falling graupel collects supercooled cloud droplets at the side which is exposed to airflow, similar to the IDEFIX conditions. It is also unlikely that the centrifugal force could cause break-off of fragile structures. Jayaratne and Grigos (1991) have found that centripetal acceleration of 9 g are needed to break off the ice structures. This leads us to the conclusion that missing rotation or random movement has no influence on the rime-splintering 370     mechanism per se.

## 4    Summary and conclusion

The Ice Droplets splintEring on FreezIng eXperiment (IDEFIX) has been designed to investigate the physical mechanisms underlying ice multiplication during riming of an ice particle falling through a cloud of supercooled droplets. In IDEFIX, the experimental conditions were selected to closely represent the environment within a mixed-phase convective cloud with 375     respect to rimer size, ambient temperature, settling velocity and droplet size range. IDEFIX was focused on understanding the potential SIP mechanisms during riming on the microscopic scale, allowing for the observation of single droplet–ice accretion events with high temporal and spatial resolution. To achieve this goal, the riming process was observed with high-speed video




microscopy and IR thermography. The detection of SI particles was carried out using a custom-build ice counter based on the inertial deposition and subsequent growth of SI particles in a supercooled sucrose solution. Therewith, SI particles with initial diameters larger than approx. 3 $\mu$m could be reliably detected.

No evidence of a productive rime-splintering SIP was found during dry and wet graupel growth, in contrast to the reports on the previous HM-type experiments, where several hundreds of SI particles per mg rime were detected at $-5\,°C$. From our observations we conclude:

– Fragmentation of droplets freezing on top of smaller accreted droplets (Griggs and Choularton, 1983) can most likely be ruled out as mechanism responsible for the effective ice multiplication during rime-splintering.

– Freezing of droplets upon glancing contact with the rimer (Mossop, 1976) was not observed.

– We found no indication of the SIP mechanisms associated with transient thermal gradient around a freezing droplet (King and Fletcher, 1976b; Dong and Hallett, 1989).

– Sublimational break-off of frail rime spires at $-7\,°C$ and $-10\,°C$ has been observed in the ice-subsaturated environment, but could not account for expected high numbers of SI particles under typical HM conditions (Hallett and Mossop, 1974).

The fact that the results from earlier HM-type experiments are not reproduced in this study can be explained in several ways. First, the number of SI particles observed in the earlier experiments could have been overestimated due to less controlled experimental conditions. This would imply that the HM SIP process is not as efficient in the mixed phase clouds as it has been assumed before. Second, the rime-splintering SIP can occur as random chain of rare burst events producing a high number of SI particles, instead of being a continuous process producing low numbers of SI particles over the whole period of riming. A different type of experiment would be required to address this issue. Finally, the SI particles produced in IDEFIX could be always significantly smaller than the detection limit of the ice counter (approx. 3 $\mu$m in diameter). As IDEFIX is operated below ice-saturation, sub-micron ice particles would evaporate completely or escape detection in the ice counter. This would imply that the actual SIP mechanism underlying the HM process could not be detected in IDEFIX. As sublimational break-off produces larger SI particles easily detectable in IDEFIX, and the spherical freezing of riming droplets and droplets freezing upon glancing contact with graupel could be excluded as potential mechanisms based on our observations, the nature of the alleged SIP mechanism behind the HM ice multiplication process remains unveiled.

In the transitional regime between dry and wet rimer growth (Schumann–Ludlam limit), pressure-induced rimer surface deformation has been observed. In analogy to droplets shattering upon freezing, such deformations could be indicative of SIP during pressure release events. Given that variation between high and low accretion rates might facilitate growth and sublimation of frail ice dendrites which could be detached upon collision with a droplet or ice particle, the role of temperature and humidity fluctuations in the clouds provides a new vantage point upon the rime-splintering ice multiplication mechanisms. At the very least, this observation points towards the possibility that rime-splintering SIP does not necessarily occur during





riming in the dry growth regime only, as has been assumed so far. However, further experiments on riming at the Schumann–Ludlam limit would be needed to asses the frequency of surface deformation occurrence and its SIP-potential.

Summarizing, the number of ice crystals detected in IDEFIX experiments is much too low to explain the rapid glaciation observed in convective and frontal clouds. It is therefore likely, that other SIP mechanisms (review given in Korolev and
415   Leisner, 2020) such as droplet shattering upon freezing, SIP due to ice-ice collisions, ice fragmentation during thermal shock, fragmentation during sublimation and activation of INP in transient super-saturation in the wake of a freezing droplet or hail have to be considered to explain the ice enhancement in mixed-phase clouds. In rapidly changing cloud conditions where no individual SIP mechanism can prevail for a long time, a combination or cascading chain of several SIP mechanisms is more likely to be the case.



**Table A1.** Impaction efficiency of PSL particles having diameters of $2\,\mu$m, $3\,\mu$m and $4.5\,\mu$m was experimentally determined for different ice counter impactor nozzle diameters and volume flows together with the corresponding theoretical particle cut-off diameter with 50% impaction efficiency based on the circular jet impaction model from Hinds (1999).

| nozzle diameter | volume flow | circular jet model $D_{p,50}$ | measured impaction efficiency of $D_{PSL}$ | | | |
|---|---|---|---|---|---|---|
| [mm] | [L min$^{-1}$] | [$\mu m$] | $1\,\mu$m | $2\,\mu$m | $3\,\mu$m | $4.5\,\mu$m |
| | 5.25 | 3.00 | 0 | 0.06 | **0.54** | 1.00 |
| 3 | 6.75 | 2.76 | | 0.16 | 0.62 | |
| | 8.24 | 2.50 | 0.08 | 0.25 | 0.66 | |
| | 5.25 | 3.95 | | 0.00 | 0.22 | 1.00 |
| 3.5 | 6.75 | 3.48 | | 0.00 | **0.52** | 1.00 |
| | 8.24 | 3.15 | | 0.34 | **0.55** | 1.00 |
| | 12.35 | 2.57 | | **0.5; 0.57** | | |
| | 6.75 | 4.21 | | | 0.27 | |
| 4 | 8.24 | 3.82 | | | 0.38 | |
| | 12.35 | 3.10 | 0.13 | 0.39 | 0.76 | 1.00 |

## Appendix A: Ice crystal detection limit

### A1 Characterization of the Ice Counter

The IC is based on the principle of a conventional impactor (Kulkarni, 2011). To determine the impaction efficiency of particles in the specifically designed IC, the theoretical particle aerodynamic cut-off diameter $D_{p,50}$ was derived by applying the circular jet model (e.g., Hinds, 1999) and verified by additional characterization experiments. Particles larger than the $D_{p,50}$ tend to impact on the substrate as their inertia causes them to escape the airflow, while particles smaller than the $D_{p,50}$ tend to follow the streamlines of the air flow. In general, $D_{p,50}$ is defined as particle diameter at which particles impact with a probability of 50% and can be described as:

$$D_{p,50} = \sqrt{\frac{9\,\eta\,W\,Stk_{50}}{\rho_p\,C_c\,U}}, \tag{A1}$$

with air viscosity $\eta$, nozzle diameter $W$, particle density $\rho_p$ and Stokes number $Stk_{50} = 0.24$ (circular jet) corresponding to $D_{p,50}$ for $500 \leq Re \leq 3000$, slip correction $C_c$ and jet velocity $U = \frac{Q}{\pi\left(\frac{W}{2}\right)^2}$, and volumetric flow rate $Q$. $C_c$ can be calculated by $C_c = 1 + \frac{\lambda_g}{d_p}\left(2.34 + 1.05\exp\left(-0.39\frac{D_p}{\lambda_g}\right)\right)$, with $\lambda_g$ the mean free path of air and particle diameter $D_p$. $D_{p,50}$ depends on the nozzle diameter $W$ with smaller nozzle diameters leading to smaller $D_{p,50}$. The IC has been constructed with the possibility to exchange different nozzle diameters (W = 3 mm, 3.5 mm, 4 mm). The theoretical description of $D_{p,50}$ is valid for particles impacting on a solid plate. For the IDEFIX IC, a viscous aqueous solution is used as the impaction surface. The air flow is directed perpendicular to the sugar solution and deforms the surface, which further complicates the theoretical description of particle impaction. To overcome this difficulty, experiments with air-suspended monodisperse polystyrene latex particles (PSL





ranging from 1 $\mu$m to 4.5 $\mu$m) were conducted to measure the impaction efficiency for this setup for different air flow rates and nozzle diameters. PSL particles are guided either through the IC or through a bypass and the corresponding particle number concentration were measured with WELAS®. Then, the impaction efficiency of the IC is determined by dividing particle

number concentration measured downstream the IC with particle number concentration of the bypass air flow. The results are summarized in Table A1. Verification experiments showed that the essential behavior is represented by the theoretical model. In detail, experimentally determined $D_{p,50}$ values are smaller than the theoretical description, resulting in a better collection efficiency for this specific setup of the IC.

**A2    Sublimation of secondary ice particles in IDEFIX**

The impaction characteristics of particles in the IC and sublimational effects limit the detectable size of SI particles. At the current configuration of IDEFIX, SI particles with 3.3 $\mu$m in diameter would shrink to 2 $\mu$m on their pathway into the IC considering the worst case scenario of saturation conditions with respect to ice (Fig. A1). With this, such particles will be impacted in the IC with an efficiency of 50 %. If SI particles have smaller diameters than 3.3 $\mu$m at their production, the probability that they will be counted within the IC decreases significantly. Therefor, this size is assumed to be the lower

detection limit.

**Appendix B:  Determination of the collision rates during riming**

**B1    Methods**

Droplet-graupel collision rates influence rimer microphysical processes and are relevant statistical parameters to describe SIP rates. Hence, they are crucial parameters for the comparability to other rime-splintering experiments. The collision rates can

be expressed in terms of mass and number of accreted supercooled droplets, respectively. For reasons of comparability, the collision rates are normalized to the graupel's geometrical cross sectional area perpendicular to the flow direction ($A_{cross}$). An average mass collision rate, also called mass accretion rate ($R_{accr,M}$) in the following, can be calculated from the frozen droplet mass accreted on the initial ice target ($\Delta M_{accr}$) in a certain time interval ($\Delta t$):

$$R_{accr,M} = \frac{1}{A_{cross}} \frac{\Delta M_{accr}}{\Delta t}. \tag{B1}$$

To derive the number accretion rate, the mass accretion rate (Eq. B1) is divided by the mean mass of the number size distribution of the accreted droplets $<M_d> = \frac{\pi}{6}\rho_w \frac{M_3}{M_0}$, with $M_0$ and $M_3$ being the 0*th* and 3*rd* moment of the distribution $n_{accr}(D)$. From additional WELAS® measurements, the airborne droplet number size distribution $n(D)$ is found to be log-normal distributed. Since the normalized size distribution of airborne droplets $n^*(D) = \frac{n(D)}{M_0} = \frac{n(D)}{\int n(D')\mathrm{d}logD'}$ is not influenced by riming, i.e. the shape of the normalized DSD does not change comparing measurements without and with an ice target

(Fig. B1), we assume $n^*(D)$ to represent the normalized accreted DSD and the following is obtained



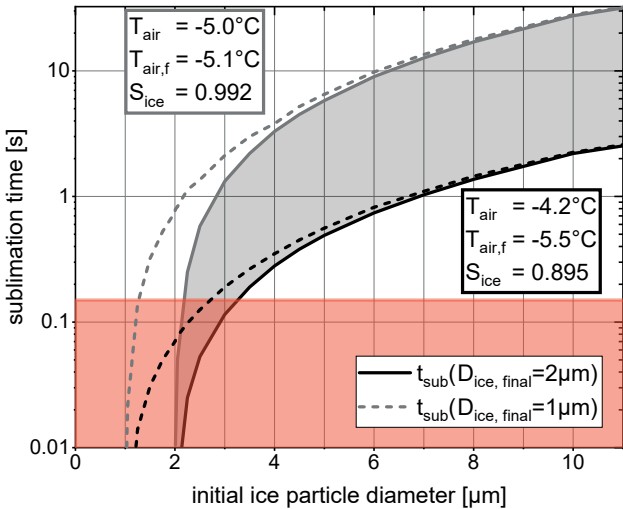

**Figure A1.** Sublimation time as a function of initial ice particle diameter. Ice particles are assumed to have a spherical shape and experience different levels of subsaturation with respect to ice. The time it takes to sublimate to a final ice particle size of $D_{\mathrm{ice,final}} = 2\,\mu\mathrm{m}$ (solid lines, light gray shaded area) and $1\,\mu\mathrm{m}$ (dashed lines) are shown since they correspond to expected IC counting efficiencies of $50\,\%$ and significant below $50\,\%$, respectively. Two extreme scenarios span the relevant parameter space regarding temperature T, frost point $T_{\mathrm{f}}$ and respective saturation with respect to to ice $S_{\mathrm{ice}}$ representative for an IDEFIX experiment, i.e., almost ice saturation (gray line) at set temperature and $90\,\%$ saturation accounting for temperature and humidity uncertainties (black line). The residence time from ice target to supercooled sugar solution surface is given (red line). When sublimation time is shorter than residence time (red area), ice particles are detected with less than $50\,\%$ for $D_{\mathrm{ice,final}} = 2\,\mu\mathrm{m}$ and might be to small to be detected for $D_{\mathrm{ice,final}} = 1\,\mu\mathrm{m}$.

$$R_{\mathrm{accr,N}} = \frac{R_{\mathrm{accr,M}}}{< M_d >} = \frac{6}{\pi \rho_{\mathrm{w}}} \frac{1}{\int D'^3 n^*(D') \mathrm{d}logD'} R_{\mathrm{accr,M}}. \tag{B2}$$

Currently, the IDEFIX setup does not allow a direct measurement of the ice target weight before and after riming. To determine the mass accretion rate for a representative set of experiments, the increase of rimer mass in a certain time interval Eq. (B1) was derived via two different and mostly independent methods using a) 2D information from HSV images, and b) the rimer surface temperature from IR thermography as input parameter for a heat balance model of graupel adopted from Pruppacher and Klett (2010).

Considering the first method, the accreted mass per time period was estimated from the projected rimer surface area as seen on the HSV images (Fig. B2). Since the ice target is attached to a cross of carbon fibers and therefor has no rotational degree of freedom, the riming occurs only on the upper part of the ice target. Neglecting microscopic rimer surface structure, the projected area of the rimer in matured stage was observed to take on a semi-ellipsoidal shape. Using the freely accessible image analysis tool imageJ, the respective half width $b_2$ and height $h_2$ are determined (Fig. B2, right). In doing so, the half ellipse was adjusted to be fully within the visible graupel area of the HSV image. The volume increase due to accretion of




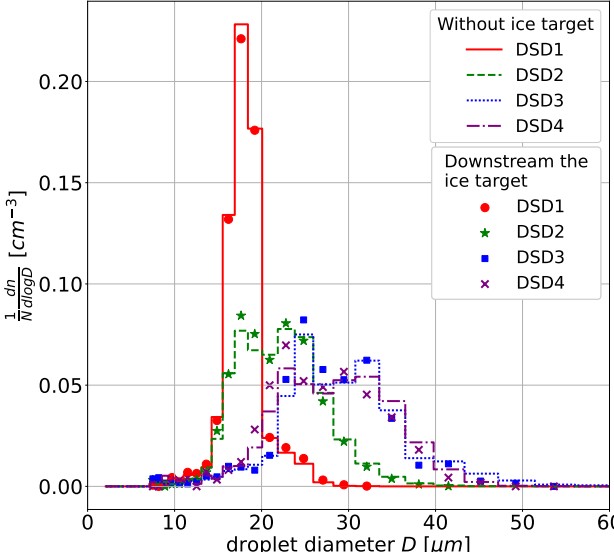

**Figure B1.** Measurements of the droplet size distributions DSD1-4 without (lines) and downstream (symbols) an ice target at -5 °C and $1\,\mathrm{m\,s^{-1}}$

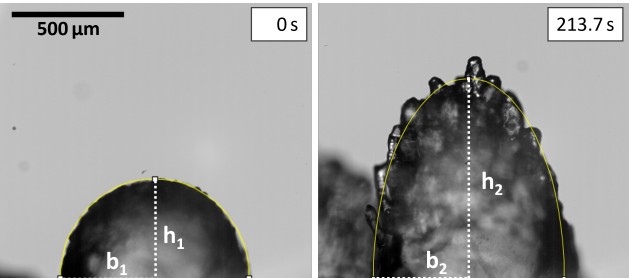

**Figure B2.** HSV-images showing a nearly spherical ice target before (left) and more ellipsoidal shaped ice target during riming (right). In this example droplets of DSD2 are accreted on the fixed ice target with an air flow velocity of $1\,\mathrm{m\,s^{-1}}$ at $-5\,$°C. The adjusted semi-ellipse of the rimer top is depicted in yellow together with the two half-axes of the semi ellipse: height $h$ and width $b$.

droplets observed between two different time steps is calculated assuming rotational symmetry and subtracting the determined ice target volumes. Following this approach, the uncertainty results either from air cavities within the rimer body or rime spire

structures that are grown beyond the ellipsoid and is assumed to be approx. 10% in volume. To obtain the accreted rimer mass from the derived volume increase, a rimer density was assumed.

    Via the second method, the mass accretion rates were calculated by applying the heat balance model of a riming particle given in Pruppacher and Klett (2010) using the rimer surface temperature from infrared thermography as input data. Assuming

steady state conditions and a constant surface temperature, the heat $q$ and mass $m$ transfer at a riming ice particle becomes




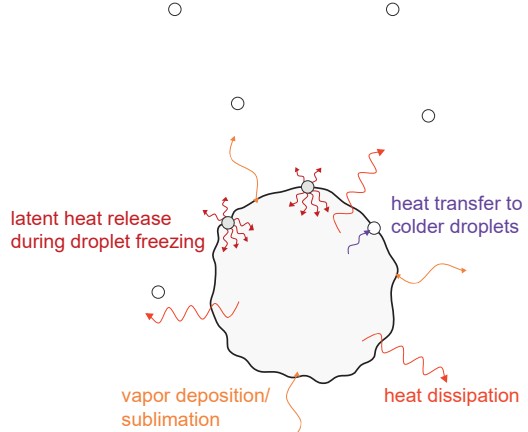

**Figure B3.** Schematics of the relevant heat transfer of a riming ice particle.

simplified to Eq. (B3) (Pruppacher and Klett, 2010, p. 680-681).

$$\left(\frac{dq}{dt}\right)_{\text{diff,h}} + \left(\frac{dq}{dt}\right)_{\text{diff,m}} + \left(\frac{dq}{dt}\right)_{\text{accr}} + \left(\frac{dq}{dt}\right)_{\text{fr}} = 0 \tag{B3}$$

The contributing terms to the riming heat balance, illustrated in Fig. B3, are (i) the heat dissipation to the environment:

$$\left(\frac{dq}{dt}\right)_{\text{diff,h}} = -\frac{A_{\text{tar}} k_{\text{a}} Nu (T_{\text{tar,s}} - T_{\text{env}})}{D_{\text{tar}}} \tag{B4}$$

with target surface area $A_{\text{tar}}$ and target diameter $D_{\text{tar}}$, thermal conductivity of dry air $k_{\text{a}}$ (param. in Beard and Pruppacher, 1971), Nusselt number $Nu = 0.88 Re^{\frac{1}{2}} Pr^{\frac{1}{3}}$ including Reynolds $Re$ and Prandtl $Pr$ number, temperature of the ice target surface $T_{\text{tar,s}}$ and of the humid air environment $T_{\text{env}}$;

(ii) heating rate due to vapor deposition or sublimation:

$$\left(\frac{dq}{dt}\right)_{\text{diff,m}} = L_{\text{s}} \left(\frac{dm}{dt}\right)_{\text{diff}} \tag{B5}$$

with latent heat of sublimation $L_{\text{s}}$ (param. in Murphy and Koop, 2005) and diffusional growth term, which can be written as

$$\left(\frac{dm}{dt}\right)_{\text{diff}} = \frac{A_{\text{tar}} D_{\text{v}} M_{\text{w}} Nu}{D_{\text{tar}} R^*} \left(\frac{e_{\text{env}}}{T_{\text{env}}} - \frac{e_{\text{tar,s}}}{T_{\text{tar,s}}}\right) \tag{B6}$$

including diffusivity of water vapor in air $D_{\text{v}}$ (param. in Hall and Pruppacher, 1976), molar weight of water $M_{\text{w}} = 0.01801\,\text{J}\,\text{kg}^{-1}$, ideal gas constant $R^* = 8.31446\,\text{J}\,\text{mol}^{-1}\,\text{K}^{-1}$, temperature and water vapor partial pressure over ice of the environment ($T_{\text{env}}$, $e_{\text{env}}$) and target surface ($T_{\text{tar,s}}$, $e_{\text{tar,s}}$), respectively;



(iii) heating rate to warm accreted droplets:

$$\left(\frac{dq}{dt}\right)_{\mathrm{accr,h}} = -c_{\mathrm{w}}(T_{\mathrm{tar}} - T_{\mathrm{env}})\left(\frac{dm}{dt}\right)_{\mathrm{accr}} \tag{B7}$$

with heat capacity of supercooled water $c_{\mathrm{w}}$ (Biddle et al., 2013) and mass accretion rate $\left(\frac{dm}{dt}\right)_{\mathrm{accr}}$

and finally, (iv) latent heat release due to freezing of accreted droplets

$$\left(\frac{dq}{dt}\right)_{\mathrm{fr}} = L_{\mathrm{f}}\left(\frac{dm}{dt}\right)_{\mathrm{accr}} \tag{B8}$$

with the latent heat of fusion $L_{\mathrm{f}}$ (Pruppacher and Klett, 2010, p.97) and the mass accretion rate $\left(\frac{dm}{dt}\right)_{\mathrm{accr}}$ to be determined,
which describes the mass increase per time interval due to accretion of colliding and subsequent freezing supercooled droplets
during riming.

Solving the rimer heat balance equation (Eq.B3) for the mass growth rate due to droplet accretion $\left(\frac{dm}{dt}\right)_{\mathrm{accr}}$ results in the
following expression:

$$\left(\frac{dm}{dt}\right)_{\mathrm{accr}} = \frac{-\left(\frac{dq}{dt}\right)_{\mathrm{diff,h}} - \left(\frac{dq}{dt}\right)_{\mathrm{diff,m}}}{L_{\mathrm{f}} - c_{\mathrm{w}}(T_{\mathrm{tar,s}} - T_{\mathrm{env}})} \tag{B9}$$

To describe the rimer surface area, the graupel was assumed to compose of a half sphere at the bottom and a rotation-
symmetric semi-ellipsoid on top. The graupel diameter and the height of the semi-ellipsoid were determined in analogy to
the first method. Applying the heat-balance model to calculate the mass accretion rate, it must be distinguished between dry
and wet growth conditions. Eq. B9 is valid for riming in the dry growth regime. During wet growth, the surface temperature
increases to the melting point enabling liquid layer formation on the rimer top and thus, $\left(\frac{dq}{dt}\right)_{\mathrm{diff,m}}$ is a function of latent heat
of evaporation instead of sublimation and the water vapor partial pressure at the target surface is considered over water instead
of over ice.

To get an idea of the scales of the terms contributing to the heat balance equation (Eq. B3), which is the basis for Eq. B9,
the individual terms were calculated for the examples of dry, wet, and transition growth presented in Fig. 4, Sect. 3.1 and are
given in Table B1.

**B2    Number and mass collision rates**

Both independent methods described in Sect. B1 were used to derive the collision rates in dependence of the DSD for a subset
of experiments at different thermodynamic conditions. The compilation of number and mass collision rates determined from
methods (a) and (b) is presented in Fig. B4.

Using 2D information from HSV images (method a), the following input parameter are applied. In addition to diameter
and height of the rimed ice target, the graupel density is needed to calculate the accreted rimer mass (Eq. B1,B2). In general,
the graupel density can vary greatly depending on formation conditions. Due to the fact, that the graupel density and thus the
volume of air cavities within the graupel structure can not be determined at the IDEFIX experiments, a density of $0.9\,\mathrm{g\,m^{-3}}$,
corresponding to compact ice without any air spaces, is assumed as an upper limit. Parameterizations of graupel density from





**Table B1.** Heat balance of a riming ice particle for three examples of dry, wet and transitional growth (Fig. 4a,b,c). The surface temperature elevation is applied as input parameter to calculate the contributions of the different heat fluxes.

| growth regime | $dT$ | $\left(\frac{dq}{dt}\right)_{\text{fr}}$ | $\left(\frac{dq}{dt}\right)_{\text{diff},h}$ | $\left(\frac{dq}{dt}\right)_{\text{diff},m}$ | $\left(\frac{dq}{dt}\right)_{\text{accr}}$ |
| --- | --- | --- | --- | --- | --- |
| | [K] | [mW] | [mW] | [mW] | [mW] |
| dry (C2_T39) | 0.6 | 0.33 | −0.21 | −0.11 | −0.00 |
| wet (C2_T56) | 4.94 | 3.34 | −1.87 | −1.32 | −0.21 |
| transition (C2_T50) | | | | | |
| dry limit | 5.03 | 7.32 | -4.62 | -2.50 | -0.50 |
| wet limit | 8.03 | 12.16 | -7.37 | -4.11 | -1.32 |

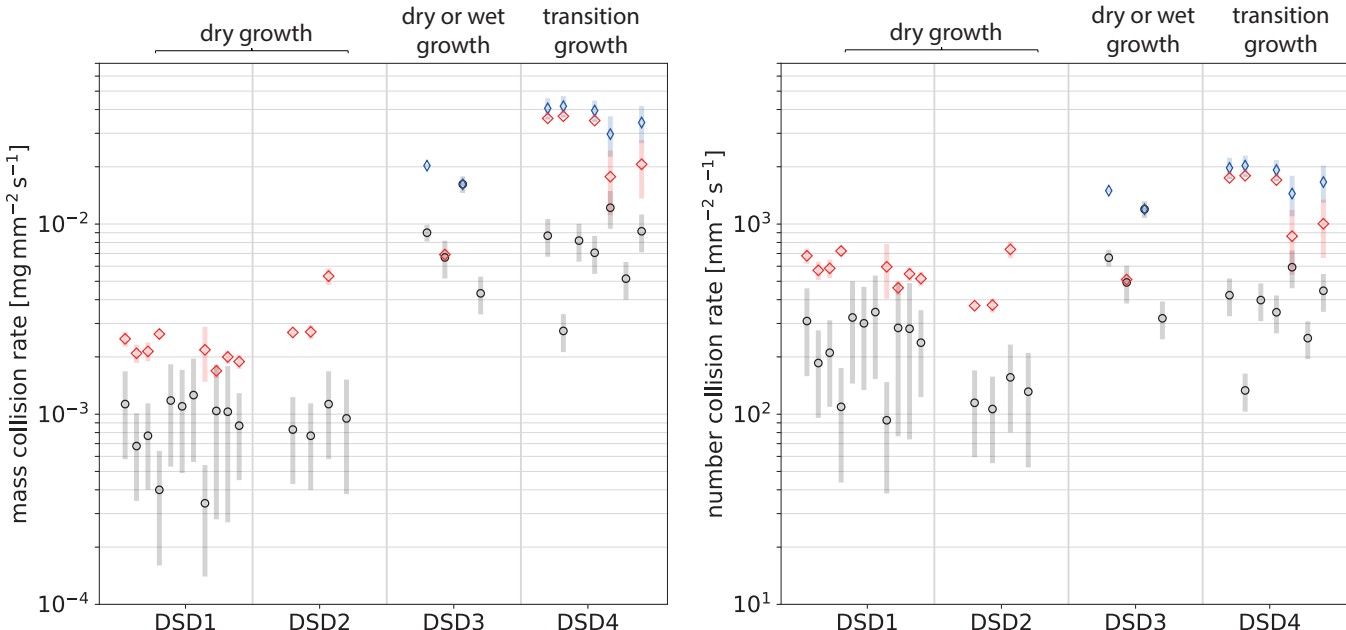

**Figure B4.** Mass and number collision rates grouped by droplet size distributions DSD1–4 and correspondingly observed growth regime. Average collision rates derived each via method (a) (black open circles) and method (b) based on the heat balance model assuming either a frozen ice target surface (dry growth conditions, red open diamonds) or a liquid water surface (wet growth conditions, blue open diamonds) depending on case. For the transition regime resulting from DSD4, the HSV-image based collision rates represent an average value accounting for the relevant ice density range (gray shaded bars), while applying the heat-balance method an upper (considering wet growth) and lower (considering dry growth) range is given.

Cober and List (1993) and Heymsfield and Pflaum (1985), where it is a function of the rimer surface temperature, the impact velocity and the median radius of the accreted cloud droplets (Macklin, 1962), are applied as lower limit. The mean values




and the possible range according to variable rimer densities are represented as symbols and gray shaded bars in Fig. B4, respectively. The number collision rates are obtained by accounting for the respective DSDs.

Applying method (b) – the determination of mass and number accretion rate derived from the heat balance model according to Pruppacher and Klett (2010) – steady state conditions can be assumed as the rimer surface temperature increases to a constant value in first approximation (see Sect. 3.1). Therefore, Eq. (B9) is easy to solve and inserted into Eq. (B1). The surface temperature elevation during riming derived from infrared thermography, target diameter and rimer surface area determined from HSV images are applied as input parameter. Thereby, it is discriminated between the dry and wet graupel growth regime by accounting for the respective properties of the ice or liquid water surface in Eq. (B9). For transitional growth (alternating ice and liquid water surface) both growth regimes are applied. Due to an inhomogeneous distribution of the surface temperature, the collision rates are calculated by using the average surface temperature displayed as symbol in Fig. B4 and the range of surface temperature variation is represented as bars.

Droplet–rimer collisions of DSD1 and DSD2 result in average number and mass collision rates of $3.6 \times 10^2 \, \mathrm{mm^{-2} \, s^{-1}}$ and $1.6 \times 10^{-3} \, \mathrm{mg \, mm^{-2} \, s^{-1}}$, respectively. For DSD3 and DSD4 the respective average number and mass collision rates are higher with $8.7 \times 10^2 \, \mathrm{mm^{-2} \, s^{-1}}$ and $1.6 \times 10^{-2} \, \mathrm{mg \, mm^{-2} \, s^{-1}}$, respectively, enabling the transition to wet growth. Comparing the approaches, method (b) results almost always in a factor of 2 to 4 times higher accretion rates than method (a) independently of the applied droplet size distribution. A possible reason for the discrepancy between the results of both methods is probably the simplified assumption of the ice target surface. However, it is remarkable that two independent approaches achieve nearly similar collision rates.

To assess the atmospheric relevant parameter space of number and mass accretions rates during riming for different cloud convection types, there are almost no atmospheric observations available. From ground-based observations and theoretical considerations, Erfani and Mitchell (2017) derived mass collision rates on the order $10^{-5} \, \mathrm{mg \, mm^{-2} \, s^{-1}}$ for frontal clouds considering liquid water contents (LWCs) from $0.05 \, \mathrm{g \, m^{-3}}$ to $0.2 \, \mathrm{g \, m^{-3}}$ and cloud droplets of median mass diameter $8 \, \mu\mathrm{m}$ and $16 \, \mu\mathrm{m}$. Jensen and Harrington (2015) simulated similar mass collision rates of $2$–$5 \times 10^{-5} \, \mathrm{mg \, mm^{-2} \, s^{-1}}$ for LWC of $0.3 \, \mathrm{g \, m^{-3}}$ and droplet mean diameter between $10 \, \mu\mathrm{m}$ and $24 \, \mu\mathrm{m}$. Since the estimated LWC and the mass accretion rates in the IDEFIX experiments are at least two orders of magnitude above this, it can be assumed that the riming conditions at IDEFIX are more representative for convective clouds.

### Appendix C: Melting point calibration of IR camera

In the experiments in which wet graupel growth could be observed with the HSV camera due to the visible formation of liquid layers on top of the rimer surface, the IR-measured maximum of the surface temperature during riming was not reaching 0 °C, but appearing between -1 °C and -2 °C. To check, whether the measured temperature maximum corresponds to the physical melting point of water, a calibration was done. Therefore, a spherical target made from epoxy resin adhesive was inserted into the IDEFIX chamber and a cap of liquid water with roughly diameter of 0.5–1 mm was generated on the top of it. Ice crystals were then inserted into the chamber by putting the tip of a screwdriver covered with ice (it was shortly dipped into



liquid nitrogen) through one of the corona needle slots into the head section of IDEFIX. Once the liquid cap was hit by an ice crystal, the freezing was initiated and the rapid spread of dendritic ice structures over the liquid could be captured with the HSV camera. From parallel surface temperature monitoring of the IR camera, we obtained a surface temperature maximum of $-1.4 \pm 0.6\,\mathrm{K}$ (average of three experiments, uncertainty range of $3\sigma$) in the moment of freezing, where the liquid cap physically should have a temperature of $0\,^\circ\mathrm{C}$.

**Appendix D: Freezing and inter-arrival time of accreted droplets on ice surface**

To calculate the individual freezing time and the inter-arrival time of two consecutive droplets hitting the same site, we consider the following system (Fig. D1): a supercooled droplet with a diameter of $D_\mathrm{d}$ and a volume of $V_\mathrm{d}$ in an air stream has the same temperature as the environment $T_\mathrm{d} = T_\mathrm{env}$. After collision with the large rimer surface, the supercooled water droplet forms a spherical liquid cap with contact angle $\theta$ at the water-air-ice interface (observation described in Sect. 3.1) and radius $a = r\sin\theta$

whereby $r$ is the radius of the imaginary sphere. During freezing, the temperature of the spherical cap rises to the melting point of water $T_\mathrm{d} = T_0 = 273.15\,\mathrm{K}$. The freezing time of the droplets can be approximated with the time of the second freezing stage, since the initial freezing and the adaptation to the ambient temperature after freezing happen on a much smaller time scale and are therefore negligible (Pruppacher and Klett, 2010; Korolev and Leisner, 2020).

The freezing time $t_2$ is determined by the transfer of latent heat to the surrounding air $q_\mathrm{cap-air}$ and the ice body $q_\mathrm{cap-ice}$.

Solving the heat balance equation for a ventilated spherical droplet cap on an ice substrate according to Macklin and Payne (1968) (cited in Pruppacher and Klett, 2010) and allowing for a variable contact angle yields

$$\alpha t_2 + \beta\sqrt{t_2} - \gamma = 0 \tag{D1}$$

with

$$\alpha = 2\pi \frac{D_\mathrm{d}\sin\theta}{(2(2+\cos\theta)(1-\cos\theta)^2)^{\frac{1}{3}}} \bar{f}[k_a(T_0 - T_\mathrm{env})$$
$$+ L_e D_v(\rho_\mathrm{v,a} - \rho_\mathrm{v,env})],$$

$$\beta = (T_0 - T_\mathrm{cap})k_\mathrm{i} \frac{D_\mathrm{d}^2(\sin\theta)^2}{(2(2+\cos\theta)(1-\cos\theta)^2)^{\frac{2}{3}}} \sqrt{\frac{\rho_\mathrm{i}c_\mathrm{i}\pi}{k_\mathrm{i}}},$$
$$\gamma = -\frac{\pi}{6}D_\mathrm{d}^3\rho_\mathrm{w}[L_\mathrm{f} - c_\mathrm{w}(T_0 - T_{env})],$$

The analytical solution for $t_2$ is

$$t_2 = \left(\frac{-\beta + \sqrt{\beta^2 - 4\alpha\gamma}}{2\alpha}\right)^2. \tag{D2}$$

Thereby $q_\mathrm{cap-air} = \alpha t_2$ and $q_\mathrm{cap-ice} = \beta\sqrt{t_2}$. $\gamma$ is the total amount of released heat. $\bar{f}$ is the mean ventilation coefficient

($\bar{f} = 1.5$), $k_\mathrm{a}$ represents the thermal conductivity of dry air (param. in Beard and Pruppacher, 1971; Pruppacher and Klett, 2010, p. 508). $k_\mathrm{i}$ is the thermal conductivity of ice (param. in Pruppacher and Klett, 2010, p. 676). $L_\mathrm{e}$ and $L_\mathrm{f}$ are the latent



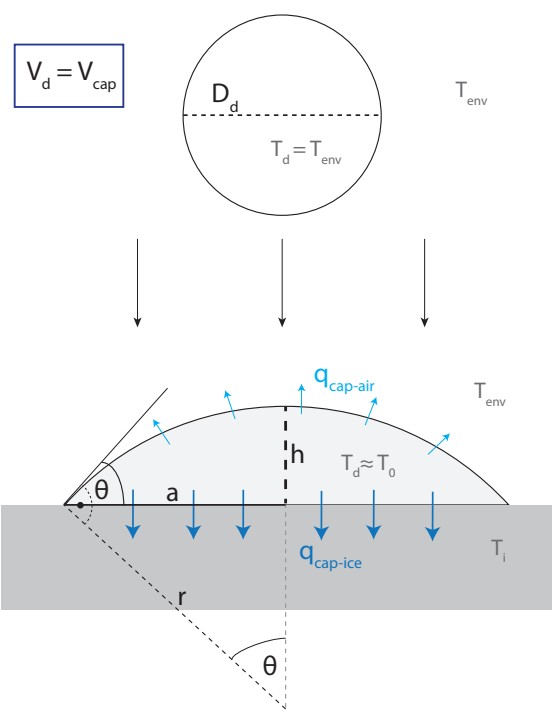

**Figure D1.** Geometry of an incident supercooled droplet ($T_d < T_0 = 273.15\,\mathrm{K}$) of diameter $D$ resulting in a freezing spherical cap with contact angle $\theta$, base radius $a$, height $h$ and radius of the imaginary sphere $r$ on an ice surface of temperature $T_i$ after collision. Heat released during the freezing process of the droplet water is transferred to the environmental air ($q_{\mathrm{cap-air}}$) of temperature $T_{\mathrm{env}}$ and to the ice surface ($q_{\mathrm{cap-ice}}$).

heats of evaporation and freezing (param. in Pruppacher and Klett, 2010, p. 97), respectively. Further, the diffusivity of water in air $D_v$ (param. in Pruppacher and Klett, 2010, p. 503), the water vapor densities of air at the cap surface $\rho_{v,a}$ and in the distant environment $\rho_{v,\mathrm{env}}$, the density of liquid water $\rho_w$ (param. in Hare and Sorensen, 1987; Pruppacher and Klett, 2010,
p. 87) and ice $\rho_i$ (param. in Baron and Willeke, 2001), the heat capacity of liquid water a $c_w$ (param. in Biddle et al., 2013) and ice $c_i$ (param. in Pruppacher and Klett, 2010, p. 87) are considered.

In addition to the individual droplet freezing time, the inter-arrival time is also relevant for the understanding of micro- and macroscopical rimer structure in the different growth regimes. Based on the number collision rate, it is possible to estimate the statistical inter-arrival time ($t_i$) between two droplets colliding at the same site on the rimer surface via Eq. D3. Both droplets
are considered to form spherical caps upon accretion with base area $A_{\mathrm{cap}}$.

$$t_i = \frac{1}{R_{\mathrm{accr,N}} A_{\mathrm{cap}}} \tag{D3}$$

The inter-arrival time is an important parameter for developing rimer surface structure during riming, as described in Sect 3.1.





**Appendix E: Rime-splintering experiments at IDEFIX**

A detailed overview of all valid rime-splintering experiments comprising the parameter space and observations of riming, SIP
and other features is given in Table E1. The few cases, where potential SI particles were detected in the IC during or after
riming, are illustrated in Fig. E1 with a pair of a HSV and IC images showing the matured rimer structure and detected ice
crystals.

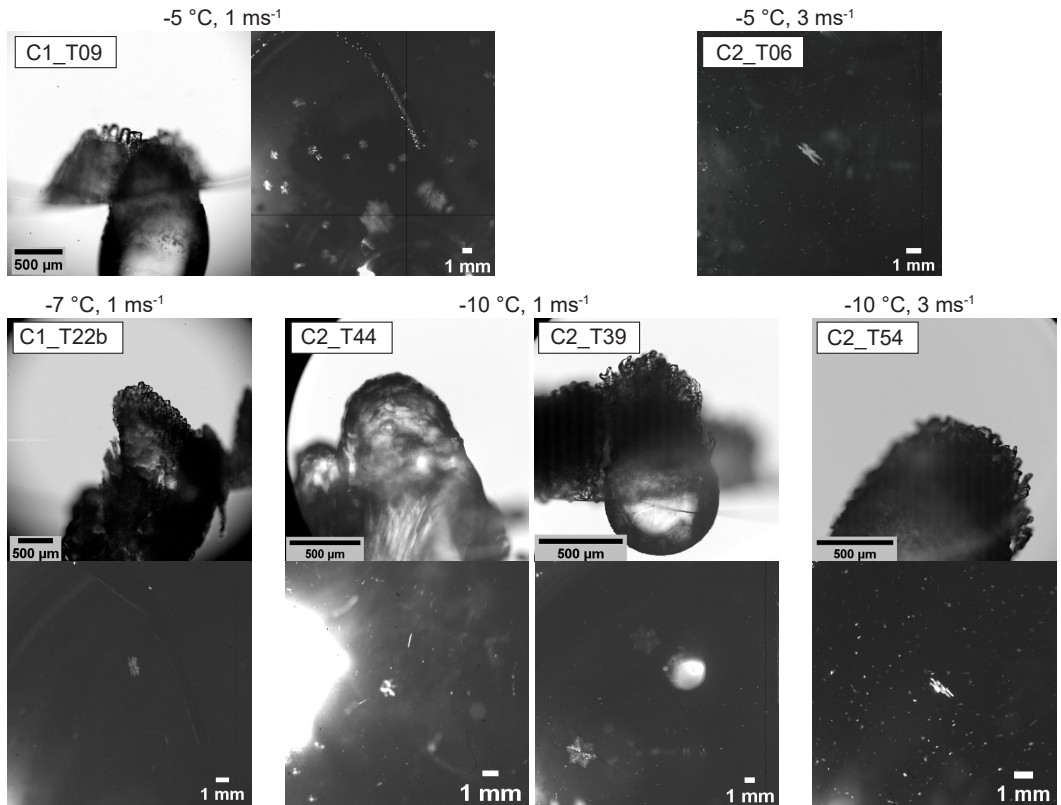

**Figure E1.** HSV images illustrating the rimed end structure of ice targets from experiments in which SIP particles have been detected with
the IC together with the corresponding enlarged cut-out of an IC image showing detected ice crystals close to the experiment end. A HSV
image of C2 T06 is missing due to lack of HSV data



**Table E1.** Overview of valid rime-splintering experiments in dependence on the main parameters: set air temperature, air velocity, droplet size distribution (DSD). The accreted mass is mainly determined by using HSV method (described in Sect. B1) or * calculated from average mass collision rate and riming time. The validation criteria comprised general functionality tests of the ice counter as well as ice counter background tests before and after each riming experiment and a minimum of approximately 0.1 mg of total accreted rime mass during the experiment. ** in this case, DSD2 was initially unstable and periods of more collisions and larger droplets occurred briefly, leading to local wetting of the ice surface.

| T [°C] | $u_{max}$ [m s$^{-1}$] | experiment | DSD | growth regime | total accreted mass [mg] | ice detected, number of crystals | ice particle occurrence, special observations |
|---|---|---|---|---|---|---|---|
| -4 | 1 | C2_T34 | DSD1 | dry | 0.1–0.2 | **no** | |
| | | C2_T38 | DSD1 | dry | 0.1* | **no** | |
| | | C2_T35 | DSD2 | dry | 0.1–0.2 | **no** | |
| | | C2_T38b | DSD3 | wet | 0.1–0.2* | **no** | |
| -5 | 1 | C1_T09 | DSD1 | dry | 0.2–0.3* | **yes, >20** | during riming |
| | | C1_T12 | DSD1 | dry | 0.1–0.2 | **no** | |
| | | C1_T13 | DSD1 | dry | 0.2–0.3 | **no** | |
| | | C1_T20 | DSD1 | dry | 0.1–0.2 | **no** | |
| | | C2_T59 | DSD1 | dry | 0.1–0.2 | **no** | |
| | | C2_T57 | DSD2 | dry | 0.1–0.2 | **no** | |
| | | C2_T56 | DSD3 | wet | 0.2 | **no** | |
| | 3 | C2_T04 | DSD1 | dry | 0.4–0.6 | **no** | |
| | | C2_T04b | DSD3 | dry | 0.1–0.8* | **no** | |
| | | C2_T06 | DSD3 | dry/transition | 0.5–1.4* | **yes, 5** | during riming |
| | 6 | C2_T60 | DSD1 | dry | 0.1 | **no** | |
| -7 | 1 | C1_T22 | DSD1 | dry | 0.2–0.4 | **no** | |
| | | C2_T15 | DSD1 | dry | 0.1 | **no** | |
| | | C2_T19 | DSD1 & DSD2 varied | dry | 0.1–0.2 | **no** | |
| | | C2_T20 | DSD1 & DSD2 varied | dry | 0.2–0.5 | **no** | |
| | 2 | C1_T22b | DSD1 | dry | 0.1–0.2 | **yes, 1** | after riming (rime spire break-off) |
| | 3 | C2_T12 | DSD1 | dry | 0.1–0.2* | **no** | |
| -10 | 1 | C2_T39 | DSD1 | dry | 0.1* | **yes, 2** | after riming |
| | | C2_T43 | DSD1 | dry | 0.1–0.3 | **no** | |
| | | C2_T40 | DSD2 | dry | 0.1* | **no** | |
| | | C2_T45 | DSD2 | dry, initially unstable** | 0.1* | **no** | ice spicule growth during riming (Fig. 11c) |
| | | C2_T44 | DSD4 | transition | 0.1–0.2 | **yes, 1** | after riming, ice spicule growth during riming (Fig. 11e) |
| | | C2_T48 | DSD4 | transition | 1.3 | **no** | ice spicule growth during riming (Fig. 11a,b) |
| | | C2_T50 | DSD4 | transition, dry | 0.4–0.5 | **no** | ascending air bubbles (Fig. 11d) |
| | 3 | C2_T54 | DSD1 | dry | 0.1–0.2 | **yes, 1** | after riming (rime spire break-off) |
| | | C2_T53 | DSD3 | dry | 0.5–0.7* | **no** | |



*Data availability.* All data can be requested from the authors. Data sets comprising the IDEFIX droplet size distributions and the overview table of valid experiments are available on Zenodo with https://doi.org/10.5281/zenodo.8405273.

*Video supplement.* HSV sequences are stored on Zenodo and can be accessed via https://doi.org/10.5281/zenodo.8405453.

*Author contributions.* JS and SH wrote the paper with contributions from all co-authors, the concept of the study was developed by SH, AK, AAK, TL and FS. The measurements and data analyis and theoretical considerations were done by JS, SH and supported by AK, AAK and FS. SH and AAK acquired the funding.

*Competing interests.* The authors declare no competing interests.

*Acknowledgements.* SH gratefully acknowledge the funding by the German Research Foundation (project number HA 8322/1-1). AK and AAK aknowldge DFG funding (project number KI 1997/1-1). AAK and TL aknowldge financial support by the Helmholtz Association under Atmosphere and Climate Programme (ATMO). AK and AAK are thankful to Stephan Vogt (IMK-AAF) for desgning the the ice counter. SH and JS are very thankful to Silvio Schmalfuß and Jens Voigtländer for supporting the model simulation to design IDEFIX, Astrid Hofmann, Bruno Wetzel and Thomas Conrath for supporting the construction, and Stephan Mertes, Dennis Niedermeier for fruitful discussions and
providing TDL and dew point instruments, respectively.



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
