# Peer review of "Secondary Ice Production - No Evidence of Efficient Rime-Splintering Mechanism"

_EGUsphere, 2023_

## Referee Comment (RC1)

**Review of "Secondary Ice Production - No Evidence of Efficient Rime-Splintering Mechanism" by Seidel, et al.**

**Overview:** Despite the significance of secondary ice production (SIP) for the formation of ice in the atmosphere, it remains one of the most mysterious microphysical processes. SIP's sensitivity to environmental conditions creates a great challenge for conducting laboratory studies on this process. Most of the laboratory investigations of SIP mechanisms were conducted from the 1960s to the 1980s. These studies yielded a broad range of differing results. The ambiguity of the outcomes of the past lab studies, in many ways, hindered the implementation of different SIP mechanisms in cloud and climate simulations.

The present work is focused on the studies of the SIP process due to the rime-splintering (Hallett-Mossop, HM) mechanism. Historically, the rime-splintering HM mechanism was considered a major SIP process, and for the last forty years, numerical simulations of clouds attributed the origin of secondary ice particles solely to the HM process. The rate of SIP due to the HM process was based on several Hallett and Mossop laboratory studies published primarily in the 1970s. The efficiency of the SIP HM process was found to be relatively high, i.e., ~300 secondary ice particles per 1mg of rime at -5C. The present study showed production rate of nearly zero secondary ice particles due to the HM process. This is an extremely important result for the cloud modeling community and for cloud physics in general.

I thoroughly reviewed the manuscript and did not find anything that would be worth criticism or modification in the existing text. The authors designed a comprehensive laboratory setup using modern technology for IR and high-speed video monitoring of the interaction of the supercooled droplet flow with the rimer. The analysis of different effects, such as collision rates, droplet freezing time, Schumann-Ludlam limit, etc., described in five Appendices, are much appreciated, and they answered many questions about the lab setup.

**Recommendation**: In my opinion, the paper can be accepted for publication as is, and due to its great importance, it should be published as soon as possible. I also sincerely hope that moving forward, the authors will continue lab studies of the HM process and, introduce rimer roation, and explore the effect of humidity on SIP.

Alexei Korolev

---

## Referee Comment (RC2)

**Review of 'Secondary Ice Production–No Evidence of Efficient Rime-Splintering Mechanism'**

Prof. Paul Connolly, University of Manchester

On the surface this appears to be an excellent piece of work to revisit and quantify the rime-splinter (RS) mechanism of secondary ice formation using up-to-date laboratory methods. I am particularly impressed by the experimental design and set-up and the paper is well-written with few formatting issues.

The finding of the paper is that the is no evidence of an efficient RS mechanism and this is summarised in Table E1 of the manuscript. Most experiments in the paper show no evidence of secondary ice particles being produced, despite the amount of accreted rime being of the order of 0.1–1 mg rime. At $-5°C$ this should have resulted in 10s–100s of secondary ice particles, but only in one experiment, at $-5°C$, were there $\sim 20$ ice particles produced.

Harris-Hobbs and Cooper [1] showed that the rates of SIP—from the trends in the early laboratory measurements of the RS mechanism—were consistent with observations to within a factor of $\sim 3$. Harris-Hobbs and Cooper developed a theory for explaining the dependence of the RS mechanism on the sizes of droplets. Mossop [2] found the production rate to depend on the presence of small ($d < 13\,\mu m$) as well as large ($d > 24\,\mu m$) droplets. This is parameterised by Harris-Hobbs and Cooper as follows:

$$P = Cf(T) \int_{D_0} g(L) \frac{\pi}{4} (L+D)^2 \times V_{impact} n(D) E(L,D) \, dD \qquad (1)$$

where $L \cong 1\text{mm}$ is the 'diameter' of the ice particle and $D$ is the diameter of the drops, with:

$$g(L) = \frac{G_{<13}}{G_{all}} \qquad (2)$$

and:

$$G_x = \int_0^x n(D) \, d^2 E(L,D) \, dD \qquad (3)$$

$C = 0.16$ and $f(T)$ represents the temperature dependence of the RS mechanism. It is unity at $-5°C$ and tapers linearly to zero at $-3°$ and $-8°C$. For the purpose of the calculations

in this review I have assumed that $E = 1$, which is a likely maximum. This means that my calculations should overestimate the splinter production rate. As the size, $L = 1$mm, refers to a single ice particle in these experiments $g(L)$ is a constant for a given droplet size distribution.

Note that in the initial Harris-Hobbs and Cooper analysis Eq 1 was a double integral, but in this analysis there is only one ice particle—we do not need to integrate of the ice particle distribution.

Furthermore, the riming rate in this analysis is:

$$R = \int n(D) \frac{\pi (L + D)^2}{4} \frac{\pi \rho_w D^3}{6} E(L, D) \times v_{impact} dD \tag{4}$$

This is the integral of the product of the PSD, the area swept out per second and the mass of the colliding drop

**Technical Issues**

It is very useful that the authors provide the droplet size distribution in Figure 2 and Table 1. I encountered some issues when trying to interpret it.

1. Figure 2: firstly, for DSD3 there is no indication of the relative amount in each of the two modes. The smaller mode looks to be more numerous, but I think this should be reported in the paper.

2. Table 1: I am not sure what equation was used to fit the parameters $D_g$ and $\sigma_g$. Is it the lognormal distribution in Eq. 5?

$$\frac{dN}{d \log D} = \frac{1}{D\sqrt{2\pi} \ln \sigma} \exp \left[ -\frac{\ln^2 (D/D_m)}{2 \ln^2 \sigma} \right] \tag{5}$$

   if this is the case then $\sigma_g$ in your table should be unit-less as it is the standard deviation of $\ln \frac{D}{D_m}$.

3. Also in Figure B1 the units of the y-axis are listed as $cm^{-3}$, but you divided by $N$ so I think it should also be unit-less.

**Analysis of size distributions**

I could not reproduce the size distributions in Figure 2 from the parameter fits in Table 1, so I decided to digitize the data using WebPlotDigitizer (`https://automeris.io/WebPlotDigitizer/`). This worked well and is shown in Figure 1.

I then tried fitting lognormal distributions to this data using the form in Eq 5. The result is shown in Figure 2. The fits approximately match the digitized data. The fit parameters are shown in Table I. As can be seen these parameters are a little bit different to yours, so I think it is worth showing the equation that was fitted to.

[Figure]

FIG. 1. Digitized data of the size distributions

TABLE I. Parameters of the size distributions from my analysis. Please note that values for $N$ are arbitrary and do not affect the calculations when scaled by riming rate.

| DSD | $N$ | $D_m$ ($\mu$m) | $\sigma_g$ |
|---|---|---|---|
| DSD1 | 100 | 18.4 | 1.08 |
| DSD2 | 100 | 20.8 | 1.28 |
| DSD3 | | | |
| mode 1 | 110 | 25.3 | 1.1 |
| mode 2 | 40 | 32.1 | 1.1 |
| DSD4 | 100 | 30 | 1.3 |

[Figure]

FIG. 2. My fits to the digitized data. Digitized data are solid lines and fit data are dashed lines.

**Harris-Hobbs and Cooper Analysis for each DSD**

I wrote a python script to calculate the SIP using the integral in Eq. 1 and the riming rate in Eq. 4 then I divided the SIP by the riming rate and divided by $1 \times 10^6$ to obtain the splinter production rate per milligram of rime accreted. The results, at $T = -5°C$, are shown in Table II. They show that the splinter production rates (2nd column of Table II) are very small for these drop size distributions, whereas if we assume all the drop sizes participate in RS the rates are much higher (3rd column of Table II). So while I am very supportive of the new set-up for studies of the RS mechanism, I think there is a major flaw in the paper to state that the measurements mean there is no evidence of the RS mechanism.

I have made the python script available to the authors, should they like to use the script I wrote. It is here `https://github.com/UoM-maul1609/dynamical-cloud-model/blob/master/pamm/python/hh_and_cooper.py`. You can alter lines 8-15 and run the script in the usual way. For the default case, the output is shown below. The important line is the last line, which is the production rate in number of splinters produced per mg of rime. To compute the rates for different size distributions alter the variable `nPSD` on line 14.

```
The integral of the PSD for n=0 is 6.2458575443444e-10

The G13 integrated over the PSD for n=0 is 1.3030461583782903e-14

The Gall integrated over the PSD for n=0 is 8.586451573231067e-09

Fraction of rime accreted of sizes less than 13 microns 1.517560714417395e-06

The riming rate integrated over the PSD for n=0 is 7.304337185264661e-16

Production rate per mg of rime 0.00040616010544473186
```

TABLE II. Parameters of the size distributions from my analysis.

| DSD | Splinters produced per milligram of rime setting $g(L) = 1$ | |
|------|:---:|:---:|
| DSD1 | $4.1 \times 10^{-4}$ | 71.3 |
| DSD2 | 0.94 | 116.9 |
| DSD3 | $5.59 \times 10^{-12}$ | 56.08 |
| DSD4 | $3.0 \times 10^{-3}$ | 31.72 |

[1] R. L. Harris-Hobbs and W. A. Cooper, Field Evidence Supporting Quantitative Predictions of Secondary Ice Production Rates, Journal of the Atmospheric Sciences **44**, 1071 (1987).

[2] S. C. Mossop, Some Factors Governing Ice Particle Multiplication in Cumulus Clouds, Journal of the Atmospheric Sciences **35**, 2033 (1978).

---

## Author Comment (AC2)

**Response to the reviewer comments by Dr. P. Connolly**

The authors would like to thank Dr. P. Connolly for giving his thoughtful feedback to our manuscript. We are very thankful for pointing our attention to the paper by Harris-Hobbs and Cooper (1987) and for providing the Python code enabling fruitful discussion of our results. To keep our response concise, we do not include here the full review provided by Dr. Connolly, which can be found at https://doi.org/10.5194/egusphere-2023-2891-RC2, but structure our response in such a way that it addresses the Dr. Connolly's review point by point. The excerpts from the reviewers' comments and criticism are highlighted red and placed in quotation marks. The changes in the manuscript will be indicated by violet color in the revised version of the manuscript.

1. The main criticism expressed by Dr. Connolly is that the experimental conditions in our study would not allow us to observe significant secondary ice production, as reported in the Hallett-Mossop type experiments. In particular, the droplets smaller than 13 µm, that are prerequisite for riming-splintering SIP (Mossop, 1985), were absent in our experiments except for one case corresponding to the droplet generating regime MGD2 and associated droplet size distribution DSD2. In support of his criticism, Dr. Connolly provided a parameterization of SIP rates based on the paper by Harris-Hobbs and Cooper (1987) and realized in the publicly available Python code (see his review). As confirmed by Dr. Connolly in the off-line private communication, there was a mistake in the original version of his code associated with the double integral in his equation (1) which led to the negligible SIP rate according to HH&C parameterization applied to the experimental conditions of our experiments. After correcting his code and introducing further refinements of the parameterization scheme (e. g., usage of diameter instead of the radius and the reduced collision efficiency between the rimer and small droplets according to Löffler and Muhr, 1972), Dr. Connolly concludes that in case of DSD2, the HH&C parameterization predicts the formation of 3 to 5 secondary ice particles with an accreted rime mass of 0.1 to 0.2 mg; while we didn't observe any secondary ice in this experiment, the number is statistically too insignificant to be considered as a proof of the absence of secondary ice production. Therefore, his main conclusion formulated in his review still holds:

"So while I am very supportive of the new set-up for studies of the RS mechanism, I think there is a major flaw in the paper to state that the measurements mean there is no evidence of the RS mechanism".

We have reproduced the calculation of Dr. Connolly and fully agree with the outcome of his numerical analysis. If the underlying physical mechanism of the HH&C parameterization is valid, and the parameterization is universally applicable to all cases of SIP involving collision of a rimer and supercooled droplets, chances of no secondary ice observation in our experimental setup would be high. However, it was not our intention to copy the experimental setup of Hallett and Mossop, with which we might even have been able to see a high number of secondarily produced ice particles. Instead, we aimed to investigate the underlying physical mechanism of potential SIP during droplet-rimer collisions under realistic conditions, especially with respect to the realistic size of a rimer particle.

We thus have to question the validity of the physical mechanism and the applicability of HH&C parameterization to all riming experiments including those that were not conducted in the H&M-type experimental setups.

The parameterization from Harris-Hobbs and Cooper (1987) is based on the observations by Mossop (1978a,b) that the SIP rate was highest when droplets smaller than 13 µm and larger than 24 µm in diameter were present in the droplet size distribution. Later, an explanation was suggested by hypothesizing that larger droplets freezing on top of smaller ones would develop spherical ice shells thus enabling pressure rise and splintering (Fig. 8b in the preprint, e.g., Choularton et al., 1978;

Choularton et al., 1980). The only direct experimental support for this hypothesis was provided by Choularton et al. (1980) where a photograph of *"…droplet of 35μm diameter, showing evidence for the disruption of a protuberance, collected at a speed of 1.5 ms and temperature -7°C"* is shown in their Figure 2c. Note however, that the droplets in this experiment were accreted on a **frost coated needle**; no accretion of smaller droplets has been reported there. The direct applicability of this evidence to the results of H&M-type experiments is, therefore, questionable.

Based on our observations, the underlying assumption of spherical freezing of larger droplets lacks physical evidence. As we state in the manuscript (page 9, line 195-202), we clearly observe all droplets spreading on smooth or rough ice surface instead of freezing spherically at T > -10°C. Our explanation of this observation is that the time required for a droplet to freeze is too long compared to the characteristic spreading time, so that the spreading cannot be arrested upon impact or midway into the freezing process. This observation, as discussed in our manuscript (page 15, line 249-252), was supported by previous studies of Dong and Hallett (1989) and Emersic and Connolly (2017), which have shown that the droplets tend to spread upon impact on smooth and rough ice surfaces at temperature above −10 °C. To the best of our knowledge, a negative control test excluding droplets smaller than 13 μm in diameter has never been performed in the HM-type experiments (e.g., Hallett and Mossop, 1974; Mossop 1976, Mossop 1978a,b; Heymsfield and Mossop, 1984; Mossop 1985; Saunders and Hosseini, 2001).

We changed and added a sentence [page 17, line 317] in the manuscript: *This is lower than the concentration ratios from 0.1 to 2.0 (or higher) used in HM-experiments, in which the efficiency of rime-splintering was found to correlate with the accretion rate of droplets smaller than 12 μm and larger than 24 μm in diameter. Different droplet size distributions were tested in the HM-type experiments (e.g. Mossop and Hallett, 1974; Mossop, 1976, 1978a, b; Heymsfield and Mossop, 1984; Mossop, 1985a; Saunders and Hosseini, 2001), but to our knowledge, a negative control test excluding droplets smaller than around 12 μm in diameter was not conducted.*

On a side note, Mossop wrote in his 1976 paper, that based on his observations, *"the formation of an ice shell around the periphery of an accreted droplet"* can be excluded [page 55, 8i]. Further, he states *"Two lines of attack appear promising. The production of splinters either from needle-like growths or by evaporation."* [page 56].

This brings us to the conclusion, that the correlation between the presence of small droplets and the high SIP rates in **the HM-type experiments** has to be based on a different physical mechanism than a hypothetical spherical freezing of larger droplets. From this follows, that the HH&C parameterization is applicable only to the experiments that closely reproduce the HM-type SIP experimental settings and cannot be used for interpretation (and even much less as an explanation) of our negative results. The detailed discussion of what mechanism it could be is beyond the scope of this response and will be addressed in the follow-up study.

We add the summary of the above discussion to the "Discussion" section of the revised manuscript [page 17, line 327-339]:

*Harris-Hobbs and Cooper (1987) presented a parameterization (HHC) relating the SI production rates and the droplet size distribution featuring droplets smaller than 13 μm and larger than 24 μm (flat DSD). This parameterization is based on the results of HM-type experiments in support of the hypothetical mechanism of spherical freezing of larger droplets landing on top of smaller ones (Choularton et al., 1978; Choularton et al., 1980). Since small droplets were present in only one of our experiments at -5°C, the HHC parameterization predicts no SIP or just a few particles for DSD2, which we, however, have not observed (see the reviewer comment by P. Connolly and our response). While we don't doubt the true nature of the correlation between the flat shape of the droplet size distribution*

*and high SIP rates observed in the HM-type experiments, we have found no evidence supporting the underlying physical mechanism. We thus conclude, that the correlation between the presence of small droplets and the high SIP rates in the HM-type experiments has to be based on a different physical mechanism, rather than a hypothetical spherical freezing of larger droplets landing on top of smaller droplets. Thus, the HHC parameterization is applicable only to the experiments that closely reproduce the HM-type SIP experimental settings and cannot be used for interpretation and even much less as an explanation of our negative results.*

**2. Technical issues**

The reviewer made us aware of some issues regarding the droplet size distributions (Table 1). To fit our droplet size distributions (DSD1-4), we used the following lognormal-fit function:

$$\frac{1}{N}\frac{dN}{dD} = \frac{1}{\sqrt{2\pi}\sigma} \exp\left[-\frac{(\ln(D)-\mu)^2}{2\sigma^2}\right] = \frac{1}{\sqrt{2\pi}\ln(\sigma_g)} \exp\left[-\frac{\ln\left(\frac{D}{D_g}\right)^2}{2\ln(\sigma_g)^2}\right]$$

with the geometric mean diameter $D_g$ and geometric standard deviation factor $\sigma_g$. As the reviewer noticed, Table 1 in the paper should contain the unitless geometric standard deviation factor together with the geometric mean (median) droplet diameter. Misleadingly, we gave the standard deviation around the arithmetic mean. Table 1 will be corrected accordingly.

|        | μ          | σ          | $D_g$ [μm]   | $\sigma_g$ [] |
|--------|------------|------------|--------------|---------------|
| DSD1   | 2.90       | 0.08       | 18.39        | 1.09          |
| DSD2   | 3.04       | 0.24       | 22.18        | 1.27          |
| DSD3   | 3.22, 3.46 | 0.08, 0.10 | 25.26, 31.8  | 1.08, 1.10    |
| DSD4   | 3.37       | 0.26       | 31.00        | 1.29          |

Further, we will mention the relative fraction of the two modes of DSD3, which is 0.7 for mode1 and 0.3 for mode2. Also, the y-label of Figure B1 is changed accordingly.

References:

Choularton, T. W., Latham, J. & Mason, B. J. Possible mechanism of ice splinter production during riming. Nature 274, 791-792, doi:10.1038/274791a0 (1978).

Choularton, T. W., Griggs, D. J., Humood, B. Y. & Latham, J. Laboratory studies of riming, and its relation to ice splinter production. Quarterly Journal of the Royal Meteorological Society 106, 367-374, doi:10.1002/qj.49710644809 (1980).

Dong, Y. Y. & Hallett, J. Droplet accretion during rime growth and the formation of secondary ice crystals. Quarterly Journal of the Royal Meteorological Society 115, 127-142, doi:10.1002/qj.49711548507 (1989).

Hallett, J. & Mossop, S. C. Production of secondary ice particles during riming process. Nature 249, 26-28, doi:10.1038/249026a0 (1974).

Harris-Hobbs, R. L. & Cooper, W. A. Field evidence supporting quantitative predictions of secondary ice production-rates. Journal of the Atmospheric Sciences 44, 1071-1082, doi:10.1175/1520-0469(1987)044<1071:fesqpo>2.0.co;2 (1987).

Heymsfield, A. J. & Mossop, S. C. Temperature-dependence of secondary ice crystal production during soft hail growth by riming. Quarterly Journal of the Royal Meteorological Society 110, 765-770 (1984).

Löffler, F. & Muhr, W. Die Abscheidung von Feststoffteilchen und Tropfen an Kreiszylindern infolge von Trägheitskräften. Chemie Ingenieur Technik 44, 510-514, doi:https://doi.org/10.1002/cite.330440804 (1972).

Mossop, S. C. Production of secondary ice particles during growth of graupel by riming. Quarterly Journal of the Royal Meteorological Society 102, 45-57, doi:10.1002/qj.49710243104 (1976).

Mossop, S. C. Influence of drop size distribution on production of secondary ice particles during graupel growth. Quarterly Journal of the Royal Meteorological Society 104, 323-330, doi:10.1002/qj.49710444007 (1978a).

Mossop, S. C. Some factors governing ice particle multiplication in cumulus clouds. Journal of the Atmospheric Sciences 35, 2033-2037, doi:10.1175/1520-0469(1978)035<2033:sfgipm>2.0.co;2 (1978b).

Mossop, S. C. Secondary ice particle-production during rime growth - the effect of drop size distribution and rimer velocity. Quarterly Journal of the Royal Meteorological Society 111, 1113-1124, doi:10.1256/smsqj.47011 (1985).